# The perfect storm? Co-occurring climate extremes in East Africa

Derrick Muheki[1], Axel A. J. Deijns[1,2], Emanuele Bevacqua[3], Gabriele Messori[4,5], Jakob Zscheischler[3,6], and Wim Thiery[1]

[1]Department of Water and Climate, Vrije Universiteit Brussel, Brussels, Belgium
[2]Department of Earth Sciences, Royal Museum for Central Africa, Tervuren, Belgium
[3]Department of Computational Hydrosystems, Helmholtz Centre for Environmental Research – UFZ, Leipzig, Germany
[4]Department of Earth Sciences and Swedish Centre for Impacts of Climate Extremes (climes), Uppsala University, Uppsala, Sweden
[5]Department of Meteorology and Bolin Centre for Climate Research, Stockholm University, Stockholm, Sweden
[6]Technische Universität Dresden, Dresden, Germany

**Correspondence:** Derrick Muheki (derrick.muheki@vub.be)

**Abstract.** Co-occurring extreme climate events exacerbate adverse impacts on humans, the economy, and the environment relative to extremes occurring in isolation. While changes in the frequency of individual extreme events have been researched extensively, changes in their interactions, dependence and joint occurrence have received far less attention, particularly in the East African region. Here, we analyse the joint occurrence of pairs of the following extremes within the same year over East Africa: river floods, droughts, heatwaves, crop failures, wildfires and tropical cyclones. We analyse their co-occurrence on a yearly timescale because some of the climate extremes we consider play out over timescales up to several months. We use bias-adjusted impact simulations under past and future climate conditions from the Inter-Sectoral Impact Model Intercomparison Project (ISIMIP). We find an increase in the area affected by pairs of these extreme events, with the strongest increases for joint heatwaves & wildfires (+940% by the end of the century under RCP6.0 relative to present day), followed by river floods & heatwaves (+900%) and river floods & wildfires (+250%). The projected increase in joint occurrences typically outweighs historical increases even under an aggressive mitigation scenario (RCP2.6). We illustrate that the changes in the joint occurrences are often driven by increases in the probability of one of the events within the pairs, for instance heatwaves. The most affected locations in the East Africa region by these co-occurring events are areas close to the River Nile and parts of the Congo basin. Our results overall highlight that co-occurring extremes will become the norm rather than the exception in East Africa, even under low-end warming scenarios.

## 1 Introduction

Climate change studies show that the frequency, intensity, and spatial extent of various extreme events have increased due to global warming (Seneviratne et al., 2021). Most of these studies report changes in individual drivers or extreme events (Frieler

et al., 2017; Lange et al., 2020). However, the extent to which these extreme events interact with each other, possibly boosting or buffering each other, has only recently started receiving attention (e.g. Batibeniz et al., 2023). At present, there is limited knowledge on the degree to which climate and impact models can be used to illustrate these interactions and dependencies (Zscheischler et al., 2020b). Interacting climate extremes are commonly termed compound climate extremes. More broadly, compound climate events are defined as a set of multiple climate drivers and/or hazards that directly affect the society and environment (Zscheischler et al., 2018, 2020a). These events often have more destructive impacts on humans, the economy, and the environment as compared to independent events (Zscheischler et al., 2018, 2020b). For example: (i) the co-occurring drought and heatwaves experienced during the 2014 California drought resulted in record-breaking water shortages and massive wildfires despite the drought itself not being the most extreme one recorded in the region (Aghakouchak et al., 2014); (ii) the co-occurring hot and dry conditions experienced in the Upper Nile Basin during the past decades resulted in crop failures and water shortages (Coffel et al., 2019; Zscheischler et al., 2020b); and (iii) the occurrence of the 2019 floods in East Africa that followed the extreme drought of 2018-2019, the floods of March–May 2018 and the drought of 2016-2017, lead to an accumulation of adverse impacts (Rateb and Hermas, 2020; FEWS-NET, 2020).

In various regions of the world, compound extremes have been experienced in the past decades (Aghakouchak et al., 2014; Rateb and Hermas, 2020; Weber et al., 2020; Witte et al., 2011; Messori and Faranda, 2023) and it is projected that associated risks will increase due to the changing climate and population exposure (de Ruiter et al., 2020). Global warming plays a major role in the increase in frequency and intensity of these compound climate extremes, resulting in more harmful climate hazards such as heatwaves, droughts, wildfires and floods (Mora et al., 2018).

According to Weber et al. (2020), East Africa (composed of Uganda, Kenya, Tanzania, Rwanda, Burundi, South Sudan, Ethiopia, Somalia, Djibouti and parts of Eritrea, Sudan, Zambia, Malawi, Mozambique, Central African Republic, and Democratic Republic of Congo (DRC)), with a population of at least 326 million (Worldometer, 2022), is the most affected region by co-occurring droughts and heatwaves in Africa, and is also highly affected by consecutive droughts and floods. The region is further expected to experience extreme heat conditions under global warming of 2°C (Harrington and Otto, 2018; Weber et al., 2020), and is prone to compound extreme events in the future (Weber et al., 2020). Although there is some historical information on the occurrence of compound climate extremes in the region (e.g., Nicholson (2014); Rateb and Hermas (2020); FEWS-NET (2020)), little is known on how often these occur (Weber et al., 2020). This indicates the need for a detailed analysis of compound extremes in East Africa, that will allow for a better understanding of the possible dependencies between extreme events, their recurrence, and the effect of different future emission scenarios on their frequency. This is not only important for disaster risk management, but it is also key for climate change adaption planning by the government authorities in the region.

The state-of-the-art climate impact simulations from the Inter-Sectoral Impact Model Intercomparison Project phase 2b (ISIMIP 2b) provide a multi-model database of climate impacts across several sectors at both global and regional scales under a range of emission scenarios (Frieler et al., 2017) and present an ideal dataset to analyze compound climate extremes. Specifically, the processed dataset of Lange et al. (2020), which contains land areas exposed to six categories of climate

extreme events on a global scale (as used in Thiery et al. (2021)), allows for the analysis of the impact of climate change on the
frequency of compound extremes across East Africa under both present-day conditions and future climate change scenarios.

In this study, we aim to understand the occurrence of compound extremes in East Africa at annual time scales, and focus specifically on co-occurring extremes. We consider the occurrence of two out of six categories of extreme events within the same year in East Africa namely: river floods, droughts, heatwaves, crop failures, wildfires, and tropical cyclones. Several of these events have been reported to directly impact the livelihood and economy of the region (Coffel et al., 2019; Jacobs et al., 2016; Zscheischler et al., 2020b; Weber et al., 2020). We explore the changes in frequency, concurrence, consecutive occurrence, and spatial extent of 15 pairs of these extreme events, and changes in their correlation. Lastly, we determine the main drivers of changes in the occurrence of co-occurring extremes in the region, by comparing an early-industrial period (1861-1910), the present day (1956-2005) and the end of the century (2050-2099) under three future climate scenarios.

## 2  Data

We use the dataset from Lange et al. (2020), who processed impact model simulations available within ISIMIP2b, and select the region 24°N and 13°S and 18°E and 55°E. The dataset has a spatial resolution of 0.5° x 0.5° and includes extreme event simulations under both historical climate conditions (1861-2005) and future climate conditions (2006-2099) following Representative Concentration Pathways (RCPs) 2.6, 6.0 and 8.5.

The dataset includes yearly maps indicating the exposed area for each of the six categories of climate extreme events as defined by Lange et al. (2020) (see Table 1). It is comprised of output from process-based impact models forced by bias-adjusted output of four Global Climate Models (GCMs) available under phase five of the Coupled Model Intercomparison Project (CMIP5), namely: GFDL-ESM2M, HadGEM2-ES, IPSL-CM5A-LR and MIROC5. This approach is followed for all the extreme event categories except for heatwaves, where the exposed land area is diagnosed directly from the GCM surface air temperature output (Thiery et al., 2021). In this study, a multi-model ensemble approach is followed such that all available impact models per extreme event category (see Table 1), driven by the four aforementioned GCMs, are used to represent the region's exposure to extremes. To guarantee physical consistency in our analyses, we only identify co-occurring extremes from cross-category impact models driven by the same GCM. For instance, in diagnosing co-occurring river floods & wildfires, we use output data from the impact model CLM45 for river floods and data from CARAIB for wildfires where both are driven by the same GCM e.g. GFDL-ESM2M. We then repeat the calculation for the same two impact models but driven both by another GCM, and so on. We finally combine the results after having computed concurrence for all GCMs. This dataset allows us to analyze the joint occurrence of extreme events within the study region for three 50-year periods: the early-industrial period (1861-1910), the present day (1956-2005) and the end of the century (2050-2099).

The dataset we use comes with several caveats. A minor caveat is that it does not contain crop failure projections under RCP8.5. More importantly, the data represents the occurrence of an extreme event category as a single event within a grid cell per year, no matter if it occurred once or several times within the same location in the same year. Finally, an extreme event such as a wildfire, river flood or tropical cyclone can only partly cover a given grid cell, whereas other extreme events (heatwaves,

droughts and crop failures) are assigned by default to the entire grid cell. Thus, for the former three extremes, we consider that a grid cell is entirely affected when more than 0.5% of the 0.5°x0.5° grid cell area is simulated to be affected by the extreme event. Whilst these are limitations of the dataset, we have three distinct motivations to use it throughout our analysis: (i) the dataset is amongst the most detailed and complete of its kind, and provides information on the occurrence of extreme events within the study region over a very long time period (from 1861 until 2099); (ii) some of the climate extremes we consider play out over longer time scales, for example droughts may last several months to even years, wildfires may rage for several months, and crop failures may result from extreme conditions during the entire growing season; (iii) the impacts of compound extremes may be larger than those for individual events even in the case where the concurrence is not on a daily timescale. These are sometimes termed temporally compounding extremes (e.g., Zscheischler et al. (2020b)). For example, impacts of drought events on vegetation can be aggravated by droughts in consecutive growing seasons (e.g., Bastos et al. (2021); Wu et al. (2022)). Similarly, societal vulnerability to floods is modulated by the occurrence of successive flood episodes (Chacowry et al., 2018), and wildfires and hydrological extremes can also compound across seasons (Yu et al., 2023; Moody and Ebel, 2012; Larsen et al., 2009). We therefore use the yearly dataset as the backbone for this study.

## 3   Methods

### 3.1   Probability of joint occurrence of extreme events

We identify co-occurring extreme events by considering the probability of joint occurrence at grid cell level (similar to Kappes et al. (2010)). The occurrence of an extreme event within a grid cell during each year is represented as a Boolean expression (Eq. 1):

$$Occurrence\ of\ an\ extreme\ event = \begin{cases} 1, & \textit{if the exposed area in a cell} \geq 0.5\% \\ 0, & \textit{otherwise} \end{cases} \tag{1}$$

We analyse the probabilities of joint occurrence of two extreme events at a single grid cell as (Eq. 2):

$$P(joint\ occurrence) = \frac{\textit{no. of years with co-occurring extremes}}{\textit{total no. of years considered}} \tag{2}$$

Here, no. of years with co-occurring extremes represents the years when occurrence of both extreme events is equal to 1. To inspect the change in area affected by these co-occurring extremes, we quantify the percentage of the study area affected by the compound event pairs (Eq. 3).

$$Percentage\ of\ area\ affected = \frac{\textit{Total area of grid cells affected by co-occurring extremes}}{\textit{Total domain area}} \cdot 100\% \tag{3}$$

**Table 1.** Definitions of extreme event categories and impact models considered in this study

| Extreme Event | Definition in Lange et al. (2020) | Impact Models |
|---|---|---|
| River floods | Daily river flow within a pixel greater than 100-year return flow during pre-industrial times | CLM45, H08, JULES-W1, LPJmL, MPI-HM, ORCHIDEE, PCR-GLOBWB, WaterGAP2 |
| Heatwaves | Occurrence in entire pixel when the Heat Wave Magnitude Index daily (HWMId) recorded that year exceeds the $99^{th}$ percentile of the HWMId during pre-industrial times. Russo et al. (2017) defines HWMId as the annual maximum magnitude of heatwaves, whereby a heatwave consists of a minimum of three consecutive days with temperatures above the daily threshold between 1981 and 2010 | HWMId99 (directly diagnosed from GCMs) |
| Droughts | Drop of soil water content below the $2.5^{th}$ percentile of the distribution during pre-industrial times considering periods longer than 6 months. Here, data on monthly soil moisture at different soil layer depths (as close as possible to 100 cm) depending on the impact model was used (See Text S2 in Lange et al. (2020)) | CLM45, H08, JULES-W1, LPJmL, MPI-HM, ORCHIDEE, PCR-GLOBWB, WaterGAP2 |
| Crop failures | Drop of crop yield below the $2.5^{th}$ percentile of the distribution during pre-industrial times | GEPIC, LPJmL, PEPIC |
| Wildfires | Total annual burnt area | CARAIB, LPJ-GUESS, LPJmL, ORCHIDEE, VISIT |
| Tropical cyclones | Exposure to hurricane-induced winds (wind speed $\geq 64$ knots) sustained for at least one minute during the year | KE-TG-meanfield |

The impact models are described in: CLM45 (Lawrence et al., 2011; Thiery et al., 2017), H08 (Hanasaki et al., 2018), JULES-W1 (Best et al., 2011), LPJmL (Schaphoff et al., 2018a, b), MPI-HM (Hagemann and Gates, 2003; Stacke and Hagemann, 2012), ORCHIDEE (Guimberteau et al., 2018), PCR-GLOBWB (Wada et al., 2014, 2016), WaterGAP2 (Müller Schmied et al., 2014, 2016), HWMId (Russo et al., 2015, 2017; Lange et al., 2020), GEPIC (Folberth et al., 2012), PEPIC (Liu et al., 2016), CARAIB (Dury et al., 2011), LPJ-GUESS (Smith et al., 2014), VISIT (Ito and Oikawa, 2002; Ito and Inatomi, 2012) and KE-TG-meanfield (Emanuel, 2013).

As already mentioned, we take a multi-impact model ensemble approach to determine the average percentage of the region affected by co-occurring extreme event pairs during each of the 50-year periods. We also calculate the maximum number of consecutive years with joint occurrence of two extreme events under historical conditions and future climate scenarios.

## 3.2 Changes in bivariate distributions

For each extreme event pair, we plot the bivariate distributions of the percentage of the region affected by each extreme event. The distributions are plotted for each time period and RCP (for the end-of-century time period). These bivariate distributions illustrate the changes in area affected by co-occurring extreme events as well as the changes in the dependence between the areas affected by individual extremes due to climate change, whereby the latter changes can significantly increase the

risks associated with co-occurring extremes (de Walle et al., 2021; Zscheischler and Seneviratne, 2017; Zscheischler et al., 2020b, 2021). This dependence is quantified with the Spearman's rank correlation coefficient, $\rho$, for the different climate scenarios (as also used by (Zscheischler and Seneviratne, 2017; Zscheischler et al., 2021)). Note that the percentage of the region affected by individual extreme event categories in the same year does not inform on whether the pairs of events occur at the same location (grid cell). Rather, it represents their joint occurrence within the entire region in that year. Thus, the bivariate distributions highlight the effects of climate change on the compound events at regional scale. In the next section, we define a metric to address the dependencies among pairs of extremes at the same location.

We also plot the distributions for individual extremes, using the Kernel density estimation (KDE) method (Węglarczyk, 2018) to calculate the probability density functions. This then allows us to analyse shifts in the percentage of area affected by either one of the extreme events in each pair.

### 3.3 Determinants of changes in co-occurring extreme occurrences

Considering that the processed impact model simulations account only for climate-induced changes in the extremes (as defined by Lange et al. (2020)), and not for other changes such as land-use, here we only analyse the climate change-driven effects on co-occurring extremes. At a given location, from a statistical perspective, the probability of co-occurring extreme events can be affected by the effect of climate change on: (i) the probability of the individual extreme events and/or (ii) the dependence between the events (Bevacqua et al., 2020; Zscheischler et al., 2020b). To gain insights into the determinants of the changes, we compute the change in the probability of co-occurring extreme events when assuming: (i) changes in the probability of extreme events in one variable only; and (ii) changes in the coupling (dependence) between the variables only (Bevacqua et al., 2020). Here, the term 'dependence' is used in a statistical sense, and does not presuppose knowledge of an underlying physical mechanism nor of causality.

We deal with binary variables (X,Y), for which the probability of joint occurrence of extreme events ($P(x,y)$) can be expressed as:

$$P(x,y) = P(x) \cdot P(y) \cdot D(x,y) \tag{4}$$

where $P(x)$ and $P(y)$ are the probability of occurrence of extreme events of X and Y, respectively, and $D(x,y)$ represents their coupling (dependence). The coupling can enhance ($D(x,y)>1$) or dampen ($D(x,y)<1$) the probability of co-occurring extremes relative to the case of independence between the variables ($D(x,y)=1$).

Here, to quantify changes in the probability of co-occurring extremes, we follow a probability ratio approach whereby the effect of climate change is determined by dividing the probability of occurrence of an event under future climate conditions by the probability of the same event under past climate (Krikken et al., 2021; Philip et al., 2022) – for the latter, we consider here the early-industrial period (1861–1910). That is:

$$PR = \frac{P(x,y)_{future}}{P(x,y)_{past}} = \frac{P(x)_{\text{future}} \cdot P(y)_{\text{future}} \cdot D(x,y)_{\text{future}}}{P(x)_{\text{past}} \cdot P(y)_{\text{past}} \cdot D(x,y)_{\text{past}}} \tag{5}$$

where the latter equality derives from Eq. 4. Based on Eq. 5, we derive changes in the probability when assuming changes in the occurrence of extremes in X only (i.e., $P(x)$), by computing the probability ratio as:

$$PR_{\text{change in X}} = \frac{P(x)_{\text{future}} \cdot P(y)_{\text{past}} \cdot D(x,y)_{\text{past}}}{P(x)_{\text{past}} \cdot P(y)_{\text{past}} \cdot D(x,y)_{\text{past}}} = \frac{P(x)_{\text{future}}}{P(x)_{\text{past}}} \tag{6}$$

where $P(x)_{\text{past}}$ and $P(x)_{\text{future}}$ can be directly estimated form the data. The change in the probability when assuming changes in $P(y)$ only is obtained similarly as $PR_{\text{change in Y}} = \frac{P(y)_{\text{future}}}{P(y)_{\text{past}}}$. We derive similarly the change in the probability of co-occurring extremes when assuming a change in the coupling only as:

$$PR_{\text{change in D}} = \frac{P(x)_{\text{past}} \cdot P(y)_{\text{past}} \cdot D(x,y)_{\text{future}}}{P(x)_{\text{past}} \cdot P(y)_{\text{past}} \cdot D(x,y)_{\text{past}}} = \frac{D(x,y)_{\text{future}}}{D(x,y)_{\text{past}}} \tag{7}$$

where $D(x,y)$ for past and future periods can be derived from Eq. 4 as $D(x,y)_{\text{past}} = \frac{P(x,y)_{\text{past}}}{P(x)_{\text{past}} \cdot P(y)_{\text{past}}}$, and $D(x,y)_{\text{future}} = \frac{P(x,y)_{\text{future}}}{P(x)_{\text{future}} \cdot P(y)_{\text{future}}}$. Eq. 7 should be interpreted carefully when changes in $P(x)$ and/or $P(y)$ are large. In fact, as a caveat of the fact that we deal with binary variables, by construction, when positive changes in $P(x)$ and/or $P(y)$ are large, the estimated future dependency tends to be small (i.e., $D(x,y)_{\text{future}} \simeq 1$) despite the continuous variables from which the binary variable X and Y possibly being coupled. This, in turn, affects the estimated $PR_{\text{change in D}}$. However, we also note that under such potentially very large changes in $P(x)$ and/or $P(y)$, such changes control the actual change in the probability of co-occurring extremes, and dependency changes become irrelevant (Bevacqua et al., 2022). In the case of very large negative changes in $P(x)$ and/or $P(y)$, the denominator in Eq. 7 would be very small, and thus it is not obvious to get a small future dependency. For a thorough assessment of the changes in the dependencies, continuous rather than binary variables X and Y (Bevacqua et al., 2020), as well as larger sample sizes (Bevacqua et al., 2023), would be required.

## 4  Results

### 4.1  Frequency and spatial extent of co-occurring extremes

An increase in the area affected by co-occurring extremes is projected for most pairs in the future climate scenarios (Fig. 1). Compound event pairs that include river floods, heatwaves, crop failures or wildfires generally show a higher increase in spatial extent compared to the compound event pairs that include tropical cyclones or droughts. This, however, does not mean that the frequency of tropical cyclones or droughts as individual extremes will not increase due to climate change, but rather means that their spatial co-occurrence with other extremes will not increase substantially across the region.

Out of the 15 pairs of co-occurring extremes (all shown in Fig. 1), 7 show substantial changes in spatial extent by the end of the century under all three RCPs; namely: (i) river floods & wildfires, (ii) river floods & heatwaves, (iii) heatwaves & wildfires, (iv) heatwaves & crop failures, (v) droughts & heatwaves, (vi) crop failures & wildfires, and (vii) heatwaves & tropical cyclones. We define substantial changes as the median of the future scenarios exceeding the 75[th] quantile of the present-day distribution. The former three pairs show the strongest relative median increase under high warming scenarios.

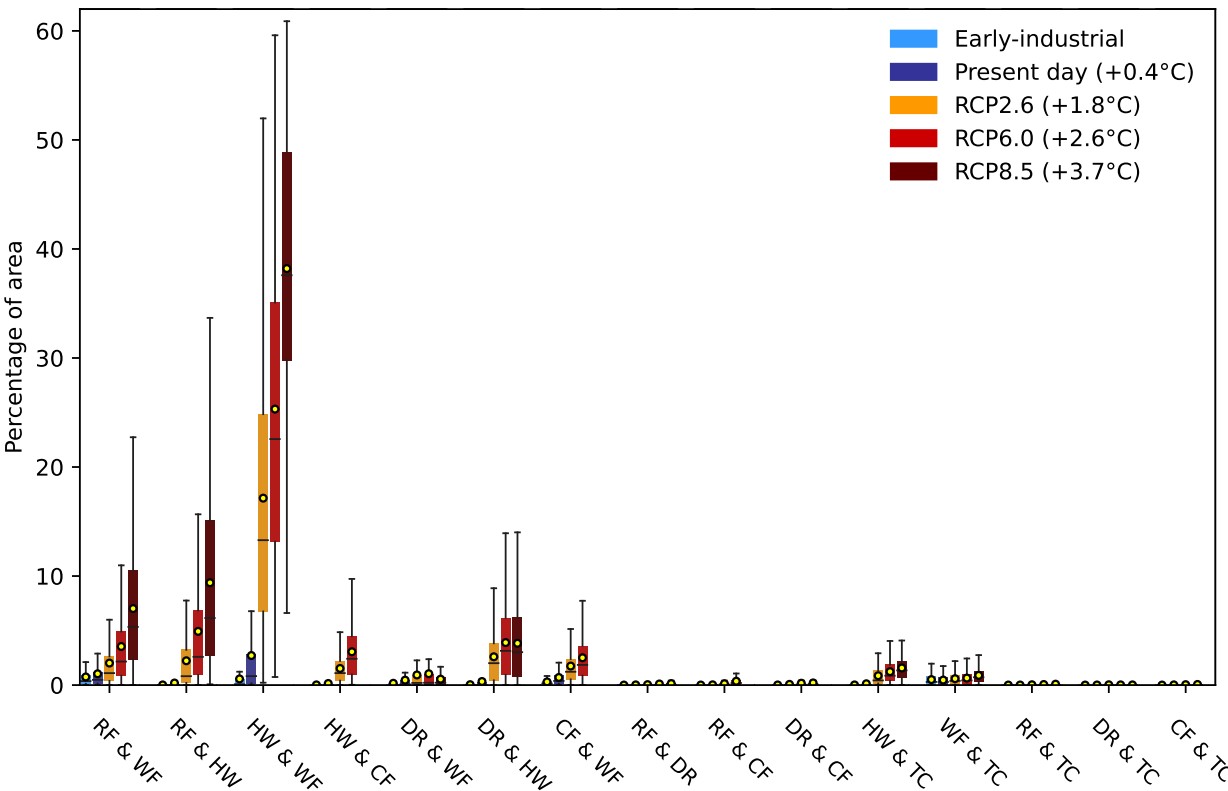

**Figure 1.** Boxplots showing the annual average percentage of the region affected by each of the 15 pairs of co-occurring extreme events under past, present and future climates. The extreme events are: RF = River floods, WF = Wildfires, HW = Heatwaves, CF = Crop failures, DR = Droughts and TC = Tropical cyclones. 50-year periods are considered for computing the average for each time window (1861-1910 for early-industrial, 1956-2005 for present day, and 2050-2099 for the future periods). A multi-model ensemble mean is shown that considers all available combinations of extreme event simulations in the dataset driven by the same GCM. Boxplots display the median (centre line) and upper and lower quartiles (box limits), with whiskers extending to the last value located within a distance of 1.5 times the interquartile range. The yellow circles show mean values. Outliers are not shown. Average global warming level (shown in the brackets within the legend) for each climate scenario, with respect to the early-industrial period, is determined using ISIMIP Global Mean Temperature (GMT) anomalies considering the mean across the respective 50-year windows.

The spatial extent of river floods & wildfires, river floods & heatwaves, and heatwaves & wildfires is projected to more than double by the end of the century under RCP6.0 compared to present-day conditions. For most of the RCP scenarios, the highest absolute increase in end-of-century spatial extent relative to the present day is projected for the heatwaves & wildfires compound event pair (+600%, +940%, and +1440% under RCP2.6, 6.0 and 8.5, respectively), followed by the river floods & heatwaves compound event pair (+400%, +900%, and +1800%) and the river floods & wildfires (+100%, +250%, and +600%).

Notable increases in the spatial extent of these three extreme event pairs can already be observed in the present-day when compared to the early-industrial period (+400%, +100%, and +100%, respectively, see Fig. 1).

We focus the rest of our analysis on the heatwaves & wildfires, river floods & heatwaves and river floods & wildfires pairs, as these are the ones displaying the largest recent and projected future changes. However we provide the results for the 12 other pairs as supplementary information (Appendix Figs. A1-A4 and Figs. B1-B4). The probability of occurrence of these co-

occurring extremes markedly increases in the end-of-century climate projections, with co-occurring river floods & heatwaves, and river floods & wildfires notably in locations close to the Nile and Congo rivers (Fig. 2c-d and 2g-h, respectively), and co-occurring heatwaves & wildfires affecting large parts of the Nile and Congo basins (Fig. 2k-l). The maps shown in the figure evidence not only an increase in affected area across the domain, but also an increase in the local event frequencies in the affected areas. All pairs show only small changes between the present-day period (1956-2005) and the early-industrial

period (1861-1910), relative to the end-of-century projected changes under the RCP2.6 scenario. In the present-day period, most locations show low values of joint probability. Higher values start to occur for co-occurring heatwaves & wildfires under RCP2.6 in specific locations across savanna ecosystems, such as southeastern DRC, and parts of the Central African Republic. Substantially higher values of joint probability of river floods & wildfires, river floods & heatwaves, and heatwaves & wildfires are widespread under RCP8.5, when compared to both present and future RCP2.6 climates. The plots clearly show a non-linear

increase in the frequency of these co-occurring events in the region under higher warming scenarios by the end of the century.

A large increase in the maximum number of consecutive years (over 50-year periods) with co-occurring heatwaves & wildfires is projected for some locations by the end of the century under RCP2.6 (Fig. 3k). Co-occurring river floods & wildfires and river floods & heatwaves instead show comparatively small increases under the same scenario (Fig. 3c, g, respectively). In contrast, a much larger increase is projected for all the co-occurring extremes under the RCP8.5 end-of-century scenario,

particularly in areas close to the Nile and Congo rivers, and along the Indian ocean coastline (Fig. 3d, h, l). For example, under RCP8.5 large parts of southern East Africa are at risk of experiencing co-occurring heatwaves and wildfires over more than 30 consecutive years in the period 2050–2099, and parts of Sudan close to the Nile may experience co-occurring river floods and heatwaves over more than 15 consecutive years.

## 4.2 Extreme event dependence

We investigate the changes in extreme event dependence for the co-occurring extremes. In a warmer climate, the bivariate distributions of the co-occurring extremes illustrate an increase in the mean (shown by the shift in marginal distribution) of affected area by river floods and heatwaves. Additionally, we also see an increase in variance (shown by the widening of the distribution) of the area affected by river floods (Fig. 4, Fig. 5a-b, and Table 2). Such changes are particularly pronounced under RCP 8.5. In contrast, the marginal distributions for area affected by wildfires show relatively small changes in the mean,

while a decrease in the variance is projected for higher warming conditions.

A small increase in dependence in co-occurring river floods & wildfires is projected by the end of the century under all scenarios, as shown by an increase in $\rho$ in these warmer future climate scenarios compared to the early-industrial conditions (Fig. 5a). Similarly, substantially higher values of $\rho$ for co-occurring river floods & heatwaves in the future climate scenarios

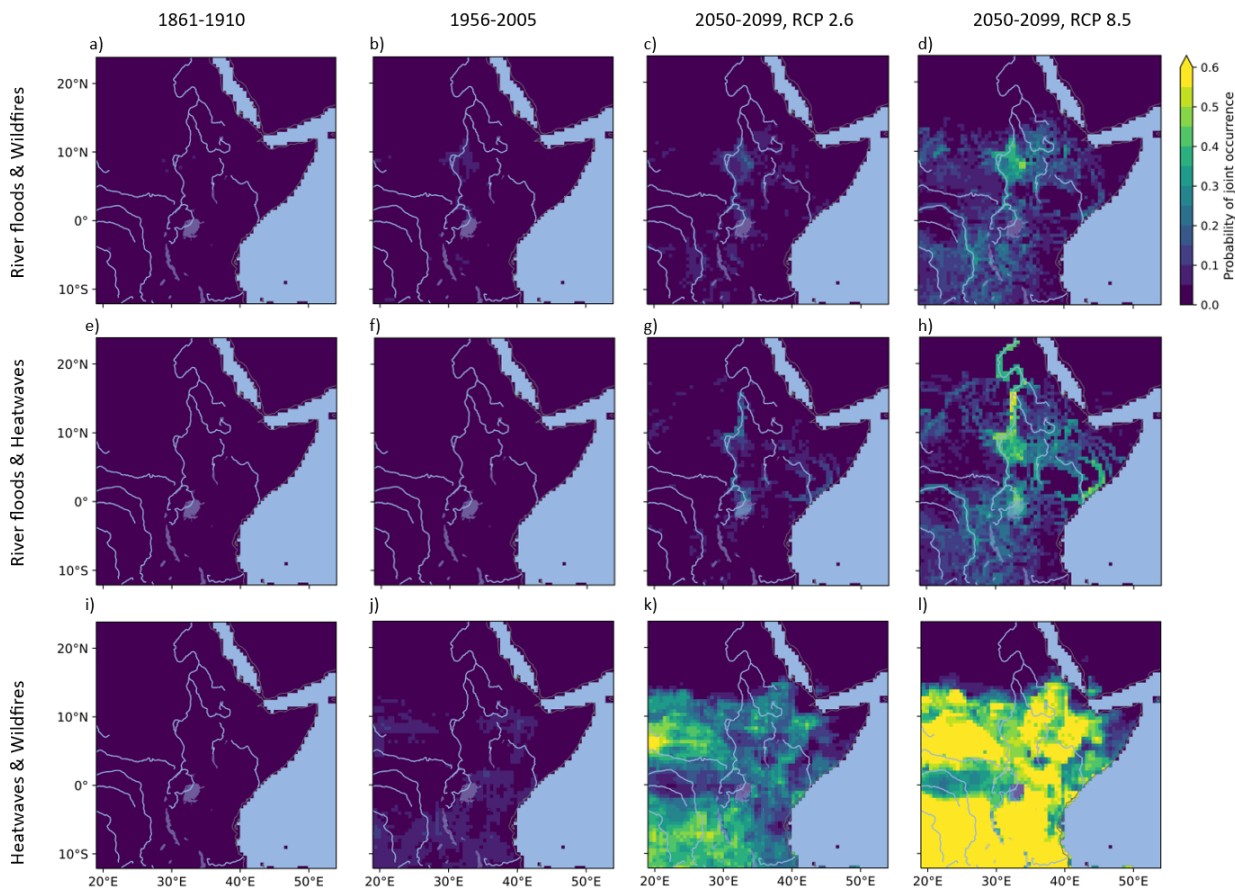

**Figure 2.** Average probability of joint occurrence of river floods & wildfires [a-d], river floods & heatwaves [e-h], and heatwaves & wildfires [i-l], across 50-year time periods (columns) representing the early-industrial period (1861-1910), the present day (1956-2005), and the end of century (2050-2099) under RCP2.6 and RCP8.5. The average probability of joint occurrence of extremes represents the multi-model ensemble mean across all available combinations of extreme event simulations in the dataset driven by the same GCM.

relative to the early-industrial period are obtained. This points to an overall increase in dependence by the end of the century,
even though it does not seem to increase monotonically with increasing radiative forcing (Fig. 5b). For co-occurring heatwaves & wildfires, the changes in $\rho$ are relatively small and a negative correlation is projected for all future climate scenarios (Fig. 5c). Additionally, we present the marginal and bivariate distributions and correlations for the other 12 co-occurring extremes in Appendix Figs. C1-C4.

### 4.3 Determinants of changes in co-occurring extremes occurrence

We next investigate the causes of the changes in the frequency of co-occurring extremes across East Africa. Here, we compare maps (e.g., Fig. 6a-c) showing the contributing probability ratios (PRs) to changes in probability of joint occurrence of extreme

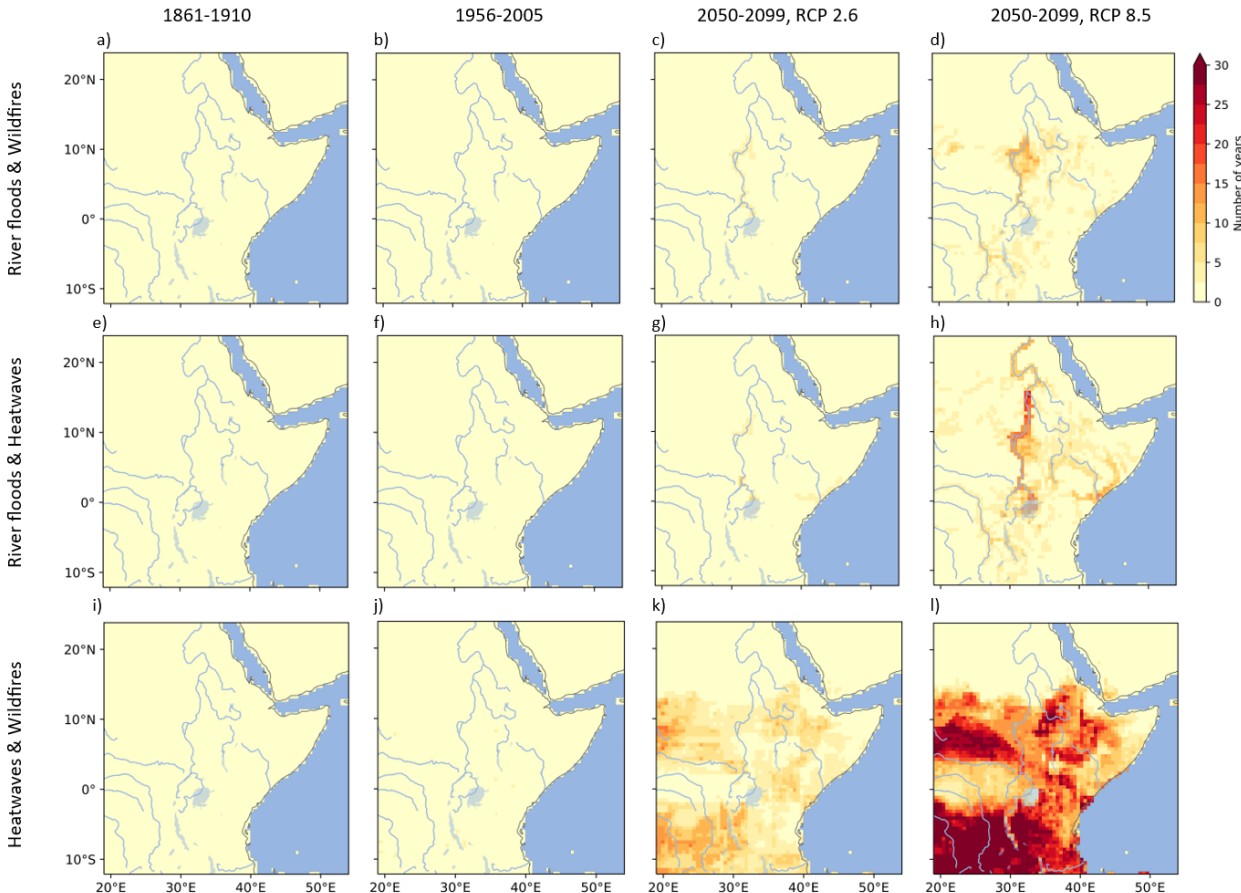

**Figure 3.** Average maximum number of consecutive years with joint occurrence of river floods & wildfires [a-d], river floods & heatwaves [e-h], and heatwaves & wildfires [i-l], across 50-year time periods (columns) representing the early-industrial period (1861-1910), the present day (1956-2005), and the end of century (2050-2099) under RCP2.6 and RCP8.5. The average maximum number of consecutive years with joint occurrence of extremes represents the multi-model ensemble mean across all available combinations of extreme event simulations in the dataset driven by the same GCM.

events under future warmer climate, whereby we: (1) assume only changes in either of the extremes per pair (first and second columns), and (2) assume changes only in the dependence of two co-occurring extremes (third column). As illustrated in Fig. 6 and Fig. D1-D4, a PR $\geq$ 1 represents more likely occurrence of the extremes and PR < 1 represents less likely occurrence, under a warmer climate in comparison to the early-industrial period.

In general, the changes in the frequency of individual extreme events control the widespread increases in compound events. This is illustrated by more locations within East Africa having higher contributing PRs (> 1) for the individual extremes per pair than the PR considering coupling (dependence) as determined using Eq. 6 and 7 respectively. Thus, changes in the coupling

**Table 2.** Mean and variance of percentage area affected by extreme events under different climate scenarios.

| Extreme Event | Mean (%) | | | | | Variance | | | | |
|---|---|---|---|---|---|---|---|---|---|---|
| | EI | PD | RCP2.6 | RCP6.0 | RCP8.5 | EI | PD | RCP2.6 | RCP6.0 | RCP8.5 |
| River floods | 1 | 2* | 4* | 7* | 11* | 3 | 5 | 19 | 37 | 72 |
| Heatwaves | 1 | 8* | 53* | 73* | 87* | 8 | 144 | 501 | 460 | 201 |
| Droughts | 2 | 4* | 4* | 5* | 4* | 5 | 14 | 14 | 17 | 1 |
| Crop failures | 0.5 | 1* | 3* | 4.1* | NA | 0.3 | 2 | 4 | 8 | NA |
| Wildfires | 40 | 39 | 39 | 40 | 48* | 275 | 267 | 245 | 265 | 103 |
| Tropical Cyclones | 1 | 1 | 2* | 2* | 2* | 1 | 1 | 1 | 1 | 1 |

Where: EI is the early-industrial period and PD is present-day conditions. Note: (1) The extreme events dataset used in this research does not contain crop failures projections under RCP8.5. (2) We considering a multi-model ensemble approach, whereby the values of mean and variance are averaged across all impact pairs driven by the same GCM. (3) The mean values denoted by an asterisk(*) represent instances where there is a statistically significant difference between the mean percentage area affected by the respective extreme event during that climate scenario, and that during the early-industrial period. Here, we use Welch's $t$ test to determine significant difference in the means (Welch, 1947).

between extremes appear to have comparably small effects (Fig. 6 and Fig. D1-D9), which is in line with previous studies (e.g., Bevacqua et al. (2020, 2022)).

The statistical determinant of the projected increase in co-occurrence of river floods and wildfires is the increase in the frequency of river floods by the end of the century under RCP8.5 (Fig. 6a-c). Similarly, the widespread strong increase in the frequency of heatwaves by the end of the century under RCP8.5 is the main determinant of the increase in probability of joint occurrence with both river floods (Fig. 6d-f) and wildfires (Fig. 6g-i). Furthermore, we illustrate the determinants of changes in occurrence of the 12 other co-occurring extremes by the end of the century under RCP6.0 and 8.5, with the former for pairs including crop failures and the latter for the rest of the co-occurring extremes (Appendix Fig. D1-D9).

## 5 Discussion

### 5.1 Frequency and spatial extent of co-occurring extremes

Three pairs of co-occurring extremes show a marked increase in their frequency and spatial extent in East Africa, under all the three future climate projections relative to the present-day climate, namely: (i) river floods & wildfires, (ii) river floods & heatwaves, and (iii) heatwaves & wildfires. These three pairs also show an increase in the maximum number of consecutive years with co-occurring extreme events, notably in areas surrounding the Nile and Congo rivers. Higher end-of-the-century warming results in an increased median frequency for all three compound event pairs. However, large differences in spatial extent of the co-occurring extremes emerge between different RCP scenarios (Fig. 1). This shows that the occurrence of these co-occurring events is highly sensitive to future global warming conditions.

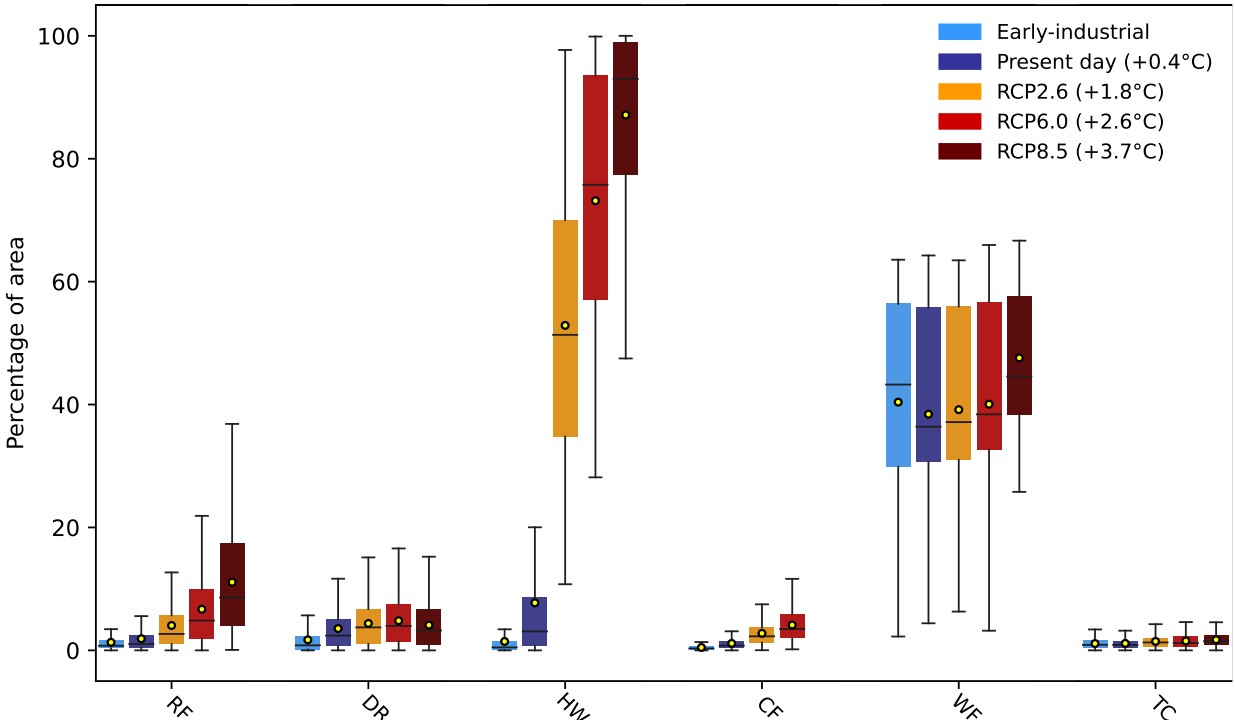

**Figure 4.** Boxplots showing the annual average percentage of the region affected by each of the six categories of extreme events under past, present and future climates. The extreme events are: RF = River floods, WF = Wildfires, HW = Heatwaves, CF = Crop failures, DR = Droughts and TC = Tropical cyclones. 50-year periods are considered for computing the average for each time window (1861-1910 for early-industrial, 1956-2005 for present day, and 2050-2099 for the future periods). A multi-model ensemble mean is shown that considers all available impact model simulations and driving GCMs in the dataset. Boxplots display the median (centre line) and upper and lower quartiles (box limits), with whiskers extending to the last value located within a distance of 1.5 times the interquartile range. The yellow circles show mean values. Outliers are not shown. Average global warming level (shown in the brackets within the legend) for each climate scenario, with respect to the early-industrial period, is determined using ISIMIP Global Mean Temperature (GMT) anomalies considering the mean across the respective 50-year windows.

Even under a low-emission RCP2.6 scenario, large increases in co-occurring extremes may be expected. For example, for co-occurring river floods & heatwaves, the percentage area affected under RCP2.6 conditions is roughly four times that affected in present-day conditions. Under RCP2.6, the percentage affected by co-occurring heatwaves & wildfires is approximately 6 times that affected in present-day conditions. To provide a term of comparison, the difference between the area affected by all three co-occurring extreme pairs discussed above under RCP2.6 and present-day conditions will be substantially larger than the difference between the present-day and early-industrial periods (Fig. 1), and is projected to be even markedly larger under the warmer RCP6.0 and 8.5 conditions. Global warming thus has a non-linear effect in increasing co-occurring extreme events

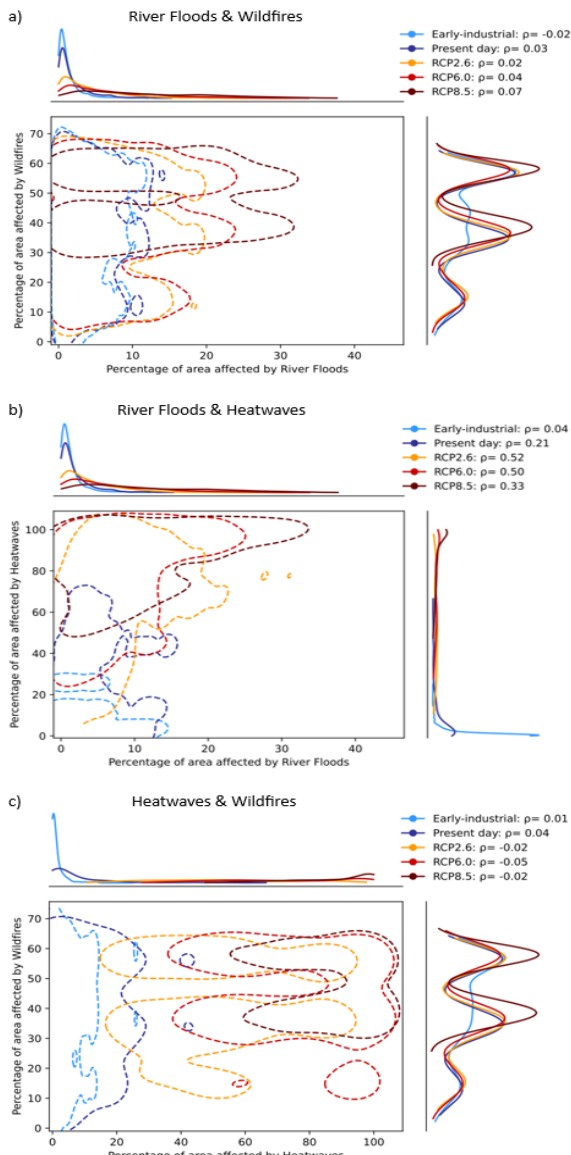

**Figure 5.** Bivariate distribution of: (a) river floods & wildfires, (b) river floods & wildfires, and (c) heatwaves & wildfires, across 50-year time periods representing the early-industrial period (1861-1910), the present day (1956-2005), and the end of century (2050-2099) under RCP2.6, 6.0, and 8.5. The marginal distributions of each extreme event (per scenario), based on the KDE method (Węglarczyk, 2018), are shown along the top and right axes of the plots. The contours (dotted lines) illustrate smooth estimates of the underlying distribution of co-occurring extremes. The 68th percentile contour, which envelopes data within one standard deviation to either side of the mean, per scenario is used to show a generalized view of the distribution of the percentage of area affected by co-occurring extremes per year during the respective scenarios.

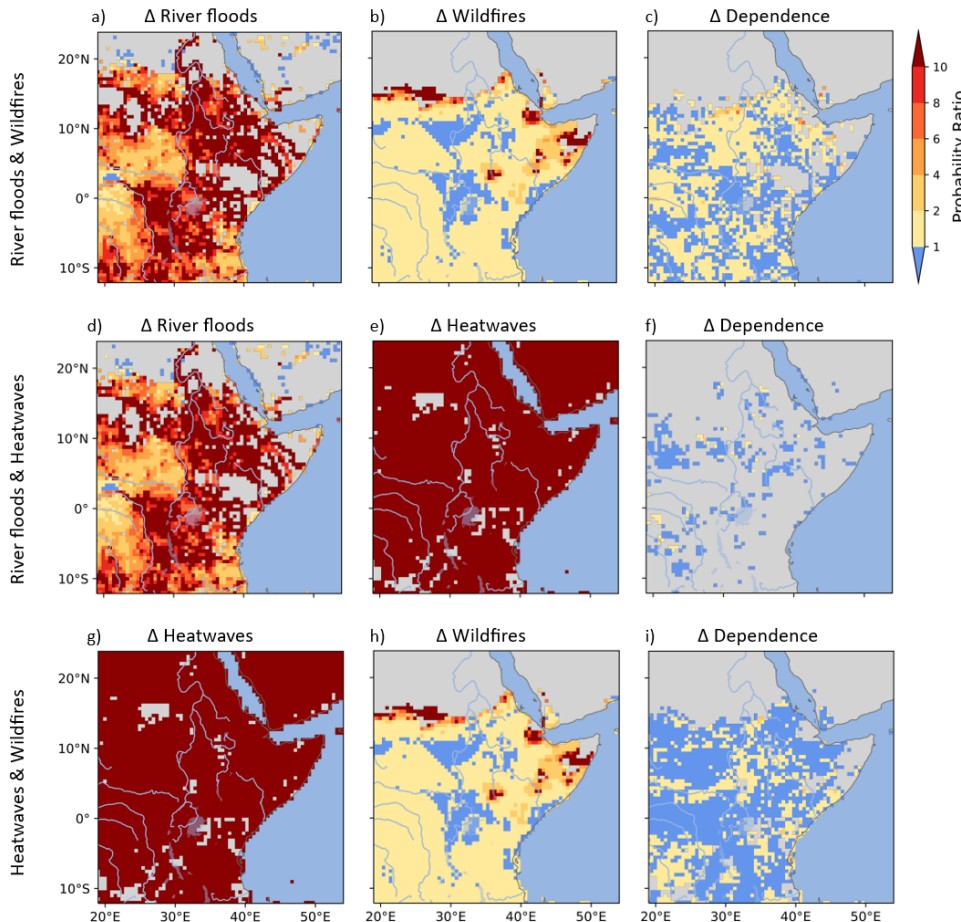

**Figure 6.** Determinants of change in co-occurring extreme event occurrence. Contributing PRs to the change in probability of joint occurrence of: river floods & wildfires [a-c], river floods & heatwaves [d-f], and heatwaves & wildfires [g-i], whereby we illustrate the PR assuming only changes in one of the extremes per pair (first and second columns) and the PR assuming changes only in the dependence of two co-occurring extremes (third column). The resulting PRs compare the end-of-century conditions under RCP8.5 to the early-industrial period conditions, whereby $PR \geq 1$ represents more likely occurrence of the extremes and $PR < 1$ represents less likely occurrence. The grey shaded areas did not experience either of the extreme events in each pair (first two columns) or co-occurrences (third column) during the early-industrial period.

across East Africa, leading to potentially larger climatic impacts in the region than may otherwise be expected. This effect is related to the shapes of the distributions of events and to how these are shifted and otherwise modified under climate change (Fig. 5).


Under all the three future climate scenarios, the hot-spots for co-occurring river floods & wildfires and river floods & heatwaves are areas in close proximity to the Nile and Congo rivers, presumably because the latter are the most frequent

source of extensive riverine flooding in the region. Perhaps less expectedly, these locations are also hot-spots for co-occurring heatwaves & wildfires under all the future scenarios, in addition to large parts of southern East Africa and the Congo basin.

Within the other 12 co-occurring extremes, relatively smaller increases in the frequency and spatial extent are projected in East Africa under RCP2.6, 6.0 and 8.5, with the highest increases expected under the warmer end-of-century scenarios for co-occurring heatwaves & crop failures, droughts & heatwaves, crop failures & wildfires, and heatwaves & tropical cyclones (Fig. 1 and Appendix Figs. A1-A4).

## 5.2    Bivariate distributions of extremes

The changes in the bivariate distributions are key for understanding the impacts of climate change on the probability of each of the extreme events in the East Africa region and their dependence. These in turn affect the frequency of co-occurrence of the extreme events (Zscheischler et al., 2020a, b). The sharp increase in mean and variance of the area affected by river floods projected under all three future climate scenarios (Fig. 4 and Table  2), in comparison to the small changes in the area affected by wildfires, explains the relatively small increase in the correlation between area affected by river floods and wildfires in a

warmer climate (Fig.  5a). This corresponds to a small increase in dependence relative to early-industrial conditions for this pair.

The percentages of area affected by river floods & heatwaves are also projected to become more correlated by the end of the century even under RCP2.6 compared to the early-industrial period, which could be a result of the sharp increase in mean and variance of the area affected by these individual extremes (Table  2, Fig. 4, and Fig. 5b). However, their correlation is projected

to decrease with the warmer future climate scenarios RCP6.0 and 8.5, relatively to RCP2.6 (Fig. 5b). This could be as result of a much stronger increase in affected area by heatwaves under RCP6.0 and 8.5, relative to the increase in area affected by river floods (Table 2 and Fig. 4).

The bivariate distributions consider the variations in affected area for each extreme event type as estimated by all possible combinations of available extreme event simulations driven by the same GCM. We discuss here the inter-model spread for

wildfires, since the percentage area affected by wildfires is the only multimodal distribution evident in our analysis. We observe that the percentage of area affected by wildfires is highly dependent on the impact model, as illustrated by the very different marginal and bivariate distributions separated by impact model (Fig. 7). Lange et al. (2020) reports that modeling uncertainty for wildfires, within this dataset, is mainly driven by the Global Vegetation Models (GVMs) whereby VISIT and ORCHIDEE simulate significantly larger burnt area in relation to the other GVMs. This could be due to differences in the representation of

human influence (such as wildfire prevention and management). It likely explains the high variation illustrated in the marginal distributions of percentage area affected by wildfires by the impact models even when driven by the same GCM, as well as the bivariate distribution of co-occurring extremes (Fig. 7). Lange et al. (2020) also discusses the uncertainties within the modelling of the other extremes which include but are not limited to: definitions of extreme events such as the return period for river floods, and representation of the carbon dioxide fertilization effect (Deryng et al., 2016) as well as agricultural management techniques

(Minoli et al., 2019) for crop failure estimation.

It is important to note that even though our results come with model uncertainties, it is unlikely that the uncertainties would alter our main findings that future climate change will greatly increase the frequency and spatial extent of co-occurring extreme events in East Africa by the end of the century for all RCP scenarios, with more drastic changes expected under higher global warming scenarios. This is illustrated by the bias-adjusted simulations that already show an increase in percentage area affected by the individual extreme events under present-day conditions in comparison to early-industrial conditions (Table 2 and Fig. 4), and project further increase under higher global warming scenarios considering the aforementioned multi-model ensemble approach. This, in turn, suggests an projected increase in their joint occurrence at regional scale.

## 5.3 Determinants of the co-occurring extremes

It is important to note that the six extreme events in this study each have different meteorological and physical drivers, i.e. heatwaves and tropical cyclones have mainly meteorological drivers, while river floods, crop failures, droughts and wildfires have mostly bio-physical drivers. Additionally, some droughts and wildfires can also be meteorology-driven. Given that we utilize extreme event data from processed impact model simulations, diving into the meteorological and physical drivers of the co-occurring extreme events presents near-insurmountable challenges. Therefore, we focus on the statistical determinants leading to co-occurring extremes in the same year in East Africa. Here, we consider the changes in the frequency of the individual extremes and their dependence per pair, under a future warmer climate scenario in comparison to the early-industrial period.

We identify heatwaves as the statistical determinant of increases in co-occurring river floods & heatwaves, and heatwaves & wildfires in East Africa by the end of the century under RCP8.5 (Fig. 6). Similarly, we identify increases in river floods as the main determinants of rising co-occurring river floods & wildfires under RCP8.5. Increases in dependence between pairs of extremes generally play a small role in explaining the modelled increases in co-occurring extremes.

As stated in the Sixth Assessment Report by Working Group II of the Intergovernmental Panel on Climate Change (IPCC), East Africa is highly likely to experience an increase in the frequency and intensity of hot days by the end of the century in comparison to the pre-industrial period, as global warming levels reach 2°C and above, with more significant increases expected at higher warming level (IPCC, 2023). As a result of the warming, more frequent heatwaves are projected by the end of the century (Niang et al., 2014; Seneviratne et al., 2021). Therefore, the increase in probability of co-occurring heatwaves & wildfires by the end of the century with heatwaves as the main determinant of the co-occurrence (Fig. 2i-l and 6g-i), can be explained by the expected warming and increase in heatwaves (with high confidence) in the region (Niang et al., 2014; Seneviratne et al., 2021).

According to Niang et al. (2014) & Seneviratne et al. (2021), the East African region is also projected to experience increased intense precipitation by the end of the century (with high confidence) under RCP8.5 scenario. This could be linked to projected changes in large-scale modes of variability, such as the Indian Ocean Dipole (IOD) and the El Niño-Southern Oscillation (ENSO), which influence precipitation across East Africa and are already showing change under present-day conditions relative to the pre-industrial period (medium confidence, Seneviratne et al., 2021). In addition to large-scale teleconnections, projected changes in mesoscale circulation and local land-atmosphere feedbacks may further affect future precipitation patterns in the

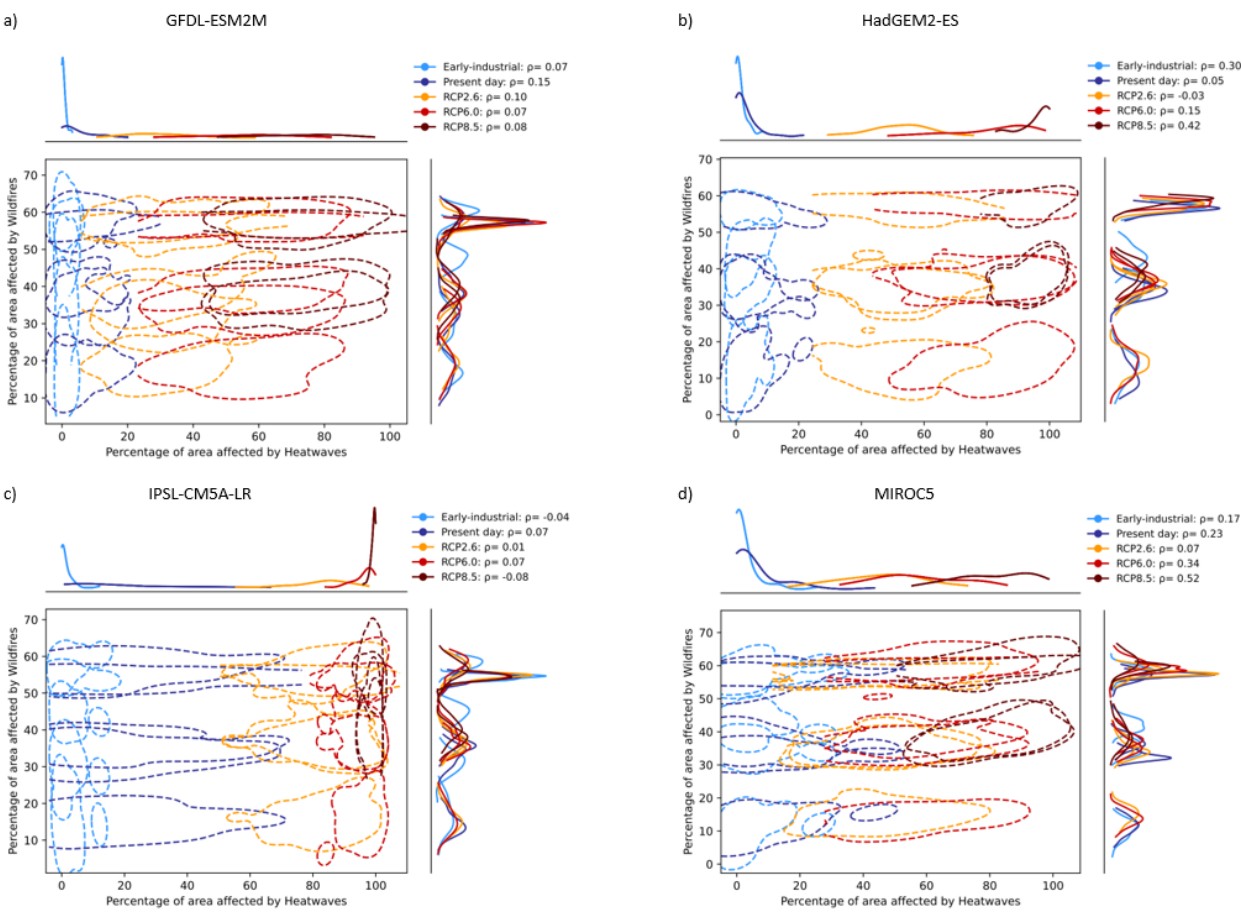

**Figure 7.** Bivariate distribution of heatwaves & wildfires across 50-year time periods representing the early-industrial period (1861-1910), the present day (1956-2005), and the end of century (2050-2099) under RCP2.6, 6.0, and 8.5. The figure separates the impact models by driving GCM: (a) GFDL-ESM2M, (b) HadGEM2-ES, (c) IPSL-CM5A-LR, (d) MIROC5. The $\rho$ values represent the average value across all the impact models under their respective historical or future climate scenarios. The marginal distributions of each extreme event (per scenario), based on the KDE method (Węglarczyk, 2018), are shown along the top and right axes of the plots. The contours (dotted lines) illustrate smooth estimates of the underlying distribution of co-occurring extremes. Here, the 68[th] percentile contour, which envelopes data within one standard deviation to either side of the mean, per scenario is used to show a generalized view of the distribution of the percentage of area affected by co-occurring extremes per year and per impact model driven during the respective scenarios. For each GCM, wildfire distributions for each impact model are shown separately.

region (Souverijns et al., 2016). However, in relation to river floods occurrence, IPCC reports that there is low confidence in the end-of-century projections of flood intensities and frequency due to inadequate data (Arias et al., 2021). Nonetheless, Alfieri et al. (2017) still projects that global warming will increase the frequency of river floods in the Nile and Congo basins, thereby

greatly affecting DRC and Sudan (Alfieri et al., 2017; IPCC, 2023). The significant increase in the frequency of co-occurring river floods & wildfires in the region by the end of the century can therefore be explained by the expected increase in frequency of river floods, which are the main determinant of co-occurrence within this pair (Fig. 2a-c and 6a-c). While, as stated above, we find that heatwaves are a major determinant for the increase in their joint occurrence with river floods by the end of the century under RCP8.5, increases in river floods themselves also shape these co-occurrences (Fig. 6d-e).

## 5.4 Potential mechanisms underlying the co-occurring extreme events

Changes in frequency of large-scale modes of variability such as the Indian Ocean Dipole (IOD) could potentially increase the coupling between the individual extreme events in East Africa. For example, the increase in frequency of extreme positive IOD events in East Africa, as a result of global warming, would lead to intense precipitation and increased susceptibility to flooding (Palmer et al., 2023). Similarly, positive IOD events substantially influence the frequency and intensity of tropical cyclones (Mahala et al., 2015), and thus under a future warmer climate, more coupling of river floods & tropical cyclones could occur during positive-IOD years by the end of the century. This aligns with our analysis that illustrates an increase in dependence (with high contributing $PR_{\text{change in D}} > 6$) in areas along the coast of East Africa under RCP8.5 (Fig. D2i). However, in general, we find that dependence of the rest of the extreme event pairs is not the main determinant of co-occurrence of extremes ($PR_{\text{change in D}} \leq 1$); instead, the increase in frequency of the individual extremes is main determinant in both present day conditions and future warmer climate (Fig. 6 and Fig. D1-D9).

## 5.5 Recommendations for further research

This research only focuses on the area exposed (in terms of pixels/grids) to co-occurring extreme events on a yearly basis. To better understand the risks associated with compound extremes, it is recommended to consider the intensity and duration of these events in this region alongside the population, assets and services exposed, as well as their vulnerability (IPCC, 2023; Zscheischler et al., 2018). Such analyses should be conducted both at sub-regional level and across the entire East African region.

Further research into co-occurring extremes in East Africa could also expand the methodology taken in this study to consider more than two extreme events occurring in the same location and year, and thus carry out a more complete multivariate analysis. Additionally, we recommend the implementation of other metrics, such as propensity (Rosenbaum and Rubin, 1983) and co-occurrence ratio (Kornhuber and Messori, 2023), to further understand the occurrence of co-occurring extremes in East Africa. Lastly, we recommend the application of our methods to other regions to illustrate how climate change may modulate co-occurring extremes in different parts of the globe.

## 6 Conclusions

This research illustrates the role of climate change in modulating the frequency and spatial extent of 15 types of co-occurring extreme events in East Africa, by considering pairs of six different categories of extreme events: river floods, droughts, heat-

waves, crop failures, wildfires, and tropical cyclones. To this end, we compare probabilities of joint occurrence, maximum number of consecutive years with co-occurring extremes, percentage of the region affected by these extremes, and their bivariate distribution during the early-industrial period, present day and the end-of-century conditions under three RCPs.

Most co-occurring extreme pairs are projected to increase in frequency and spatial extent by the end of the century due to climate change, with co-occurring extremes involving river floods, heatwaves, or wildfires projected to have the largest increases in East Africa. Increases in heatwaves and river floods are identified as the main determinants of the changes in frequency and spatial extent of the above co-occurring extremes.

For most of the co-occurring extreme event pairs, an increase in probability of joint occurrence of the extreme events is found already in the present day when compared to the early-industrial period. These changes are projected to substantially amplify across the East Africa region by the end of the century. Notably, the effects of climate change appear to be nonlinear, meaning that higher emission scenarios amplify disproportionately the frequency of several pairs of co-occurring extremes.

Our results, in conclusion, endorse the need for governments on both a regional and global scale to set policies and long-term goals in alignment with the Paris Agreement to limit global average warming to well below 2°C above pre-industrial levels as a means to reduce the risks and impacts associated with climate change (UNFCCC, 2016). Rapid, sustained and deep greenhouse gas emission reductions by the governments worldwide could significantly reduce the risks associated with co-occurring climate extremes in East Africa. We nonetheless underscore that large increases in co-occurring events in East Africa may occur even under low-emission scenarios. Additionally, governments and local authorities in East Africa should urgently embark on climate change adaptation measures to reduce the risk associated with the upcoming escalation of co-occurring extremes in the region.

*Code and data availability.* The postprocessed ISIMIP2b dataset used is available on Zenodo at https://zenodo.org/record/5497633. Correspondence and additional requests should be addressed to Derrick Muheki (derrick.muheki@vub.be). All scripts used for the analyses are available through the GitHub repository of the Department of Water and Climate at the Vrije Universiteit Brussel (https://github.com/VUB-HYDR/co-occurring_climate_extremes_in_east_africa).

## Appendix A: Probability of Joint occurrence

In most of the co-occurring extreme event pairs, no substantial increase in the probability of joint occurrence of the extreme events within the region is already observed in the present day when compared to the early-industrial period (Figures A1 - A4). 390 For some extreme event pairs, these probabilities increase substantially in both spatial extent and frequency across the region by the end-of-century considering future climate projections under the low RCP2.6 and under RCP6.0 and 8.5.

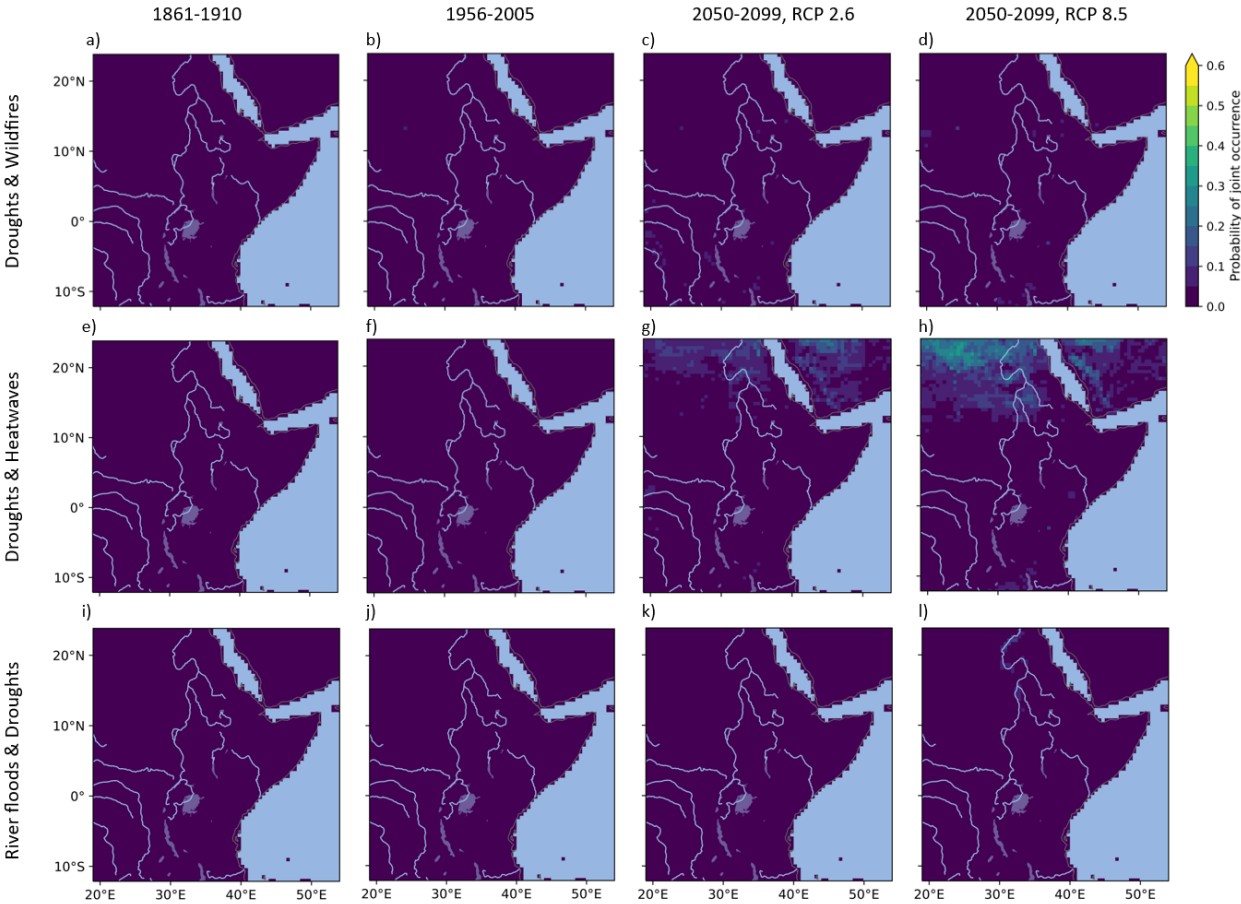

**Figure A1.** Average probability of joint occurrence of droughts & wildfires [a-d], droughts & heatwaves [e-h], and river floods & droughts [i-l], across 50-year time periods (columns) representing the early-industrial period (1861-1910), the present day (1956-2005), and the end of century (2050-2099) under RCP2.6 and RCP8.5. The average probability of joint occurrence of extremes represents the multi-model ensemble mean across all available combinations of extreme event simulations in the dataset driven by the same GCM

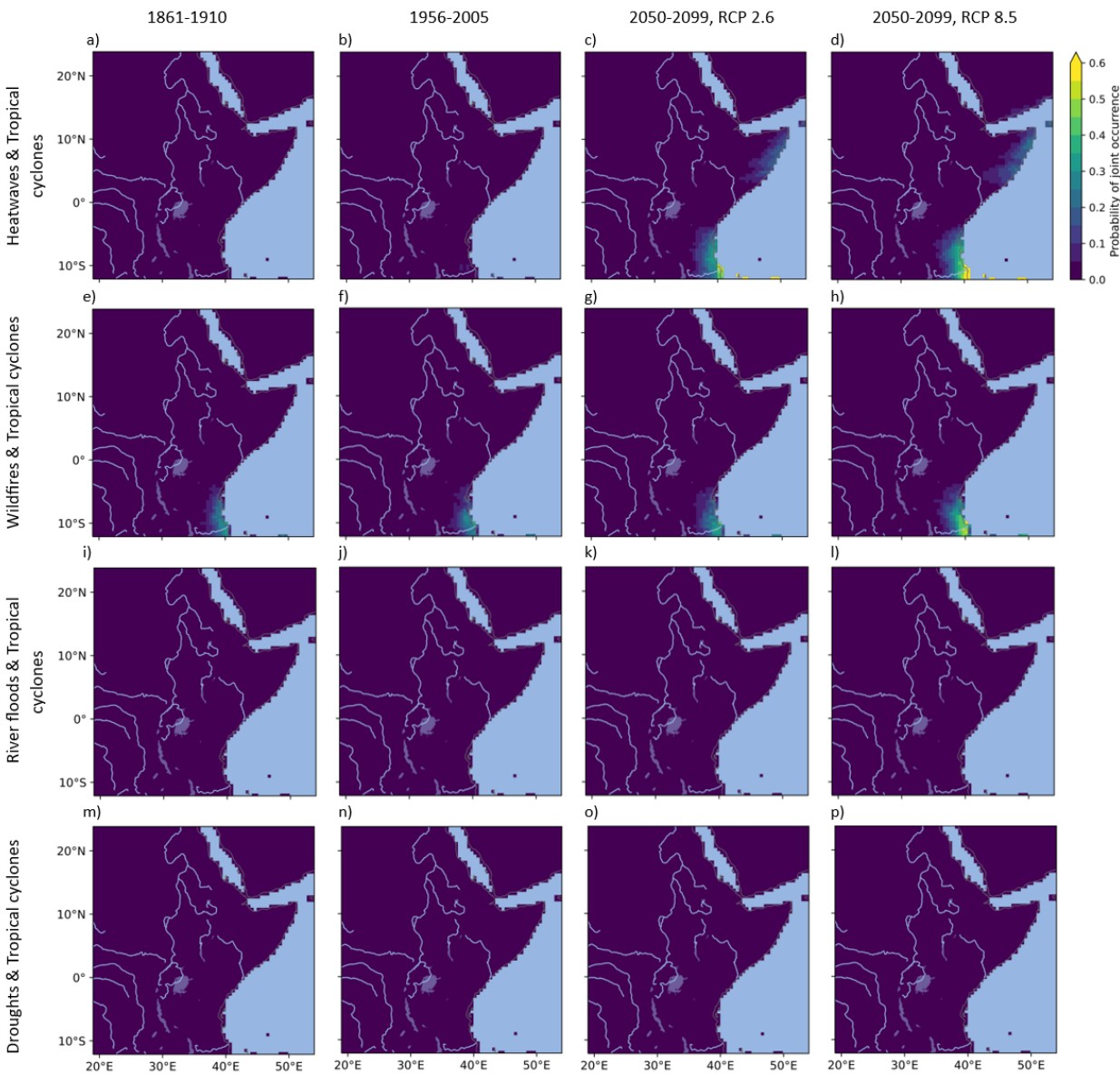

**Figure A2.** Average probability of joint occurrence of heatwaves & tropical cyclones [a-d], wildfires & tropical cylones [e-h], river floods & tropical cylones [i-l], and droughts & tropical cyclones [m-o], across 50-year time periods (columns) representing the early-industrial period (1861-1910), the present day (1956-2005), and the end of century (2050-2099) under RCP2.6 and RCP8.5. The average probability of joint occurrence of extremes represents the multi-model ensemble mean across all available combinations of extreme event simulations in the dataset driven by the same GCM

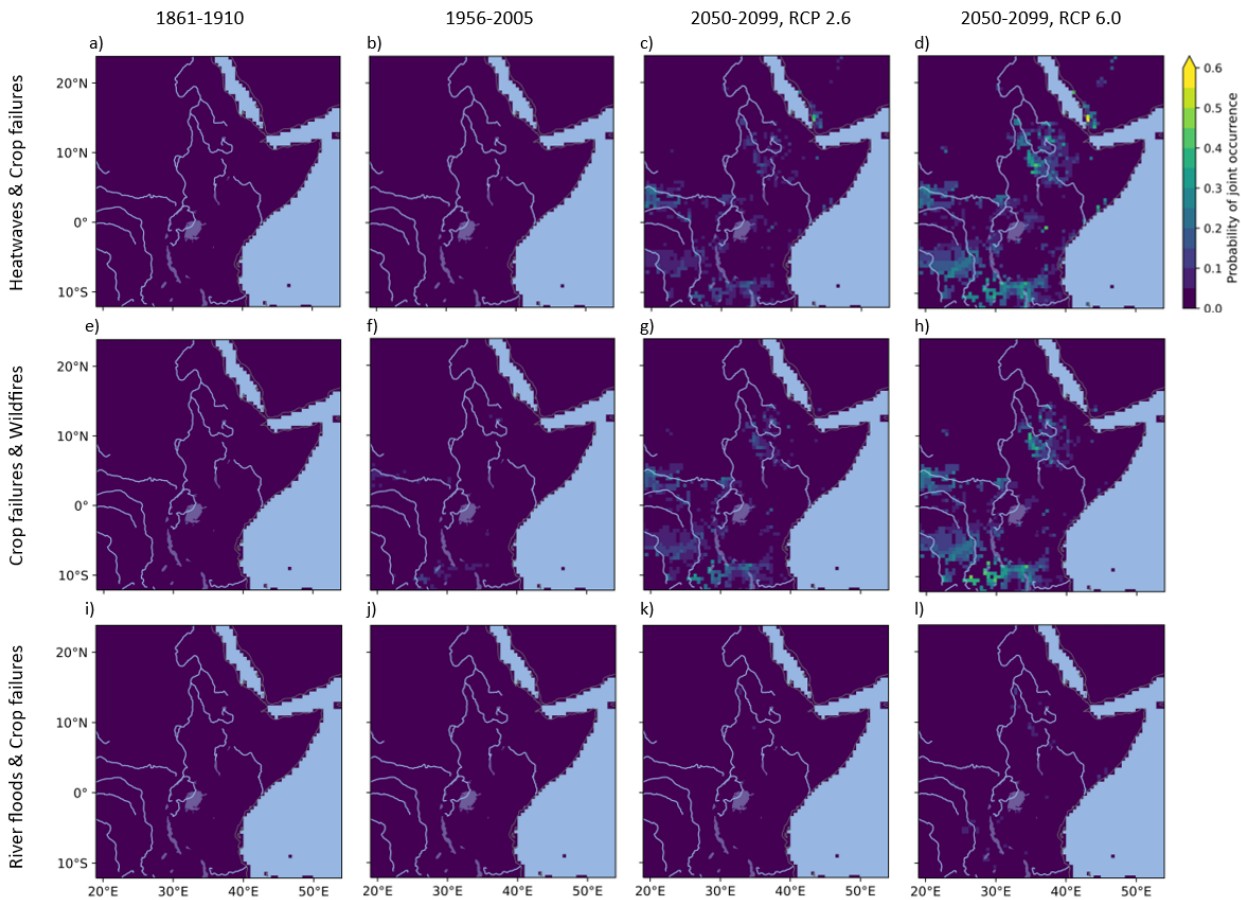

**Figure A3.** Average probability of joint occurrence of heatwaves & crop failures [a-d], crop failures & wildfires [e-h], and river floods & crop failures [i-l], across 50-year time periods (columns) representing the early-industrial period (1861-1910), the present day (1956-2005), and the end of century (2050-2099) under RCP2.6 and RCP6.0. The average probability of joint occurrence of extremes represents the multi-model ensemble mean across all available combinations of extreme event simulations in the dataset driven by the same GCM

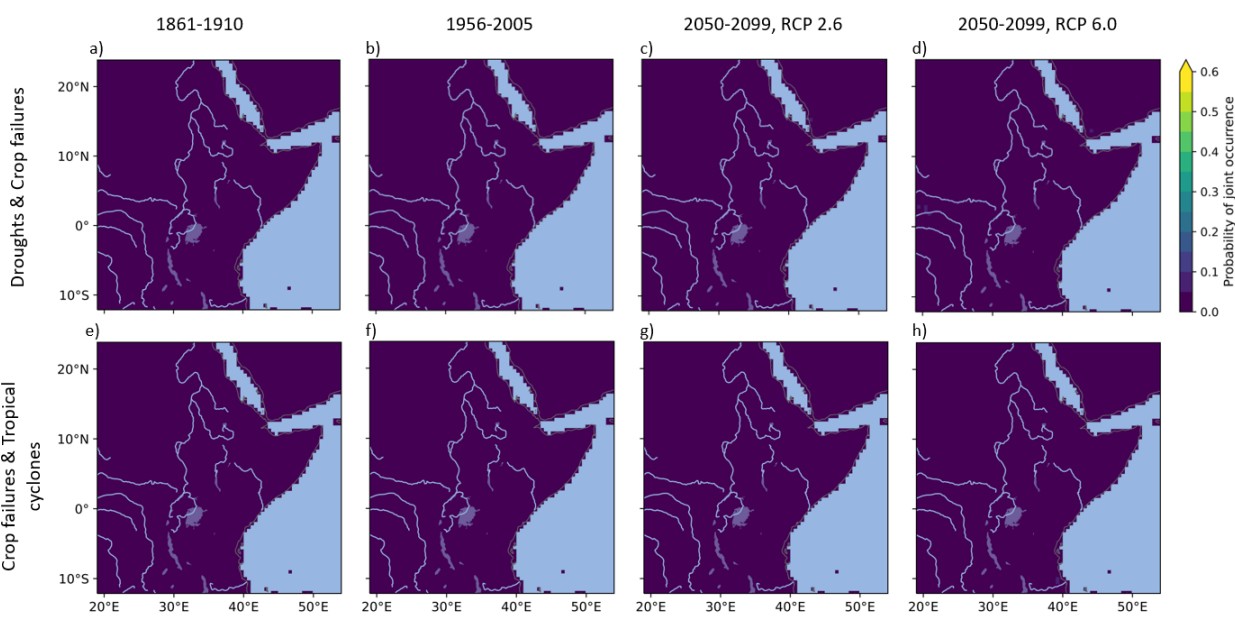

**Figure A4.** Average probability of joint occurrence of droughts & crop failures [a-d], and crop failures & tropical cyclones [e-h], across 50-year time periods (columns) representing the early-industrial period (1861-1910), the present day (1956-2005), and the end of century (2050-2099) under RCP2.6 and RCP6.0. The average probability of joint occurrence of extremes represents the multi-model ensemble mean across all available combinations of extreme event simulations in the dataset driven by the same GCM

## Appendix B: Average maximum number of consecutive years with compound extremes

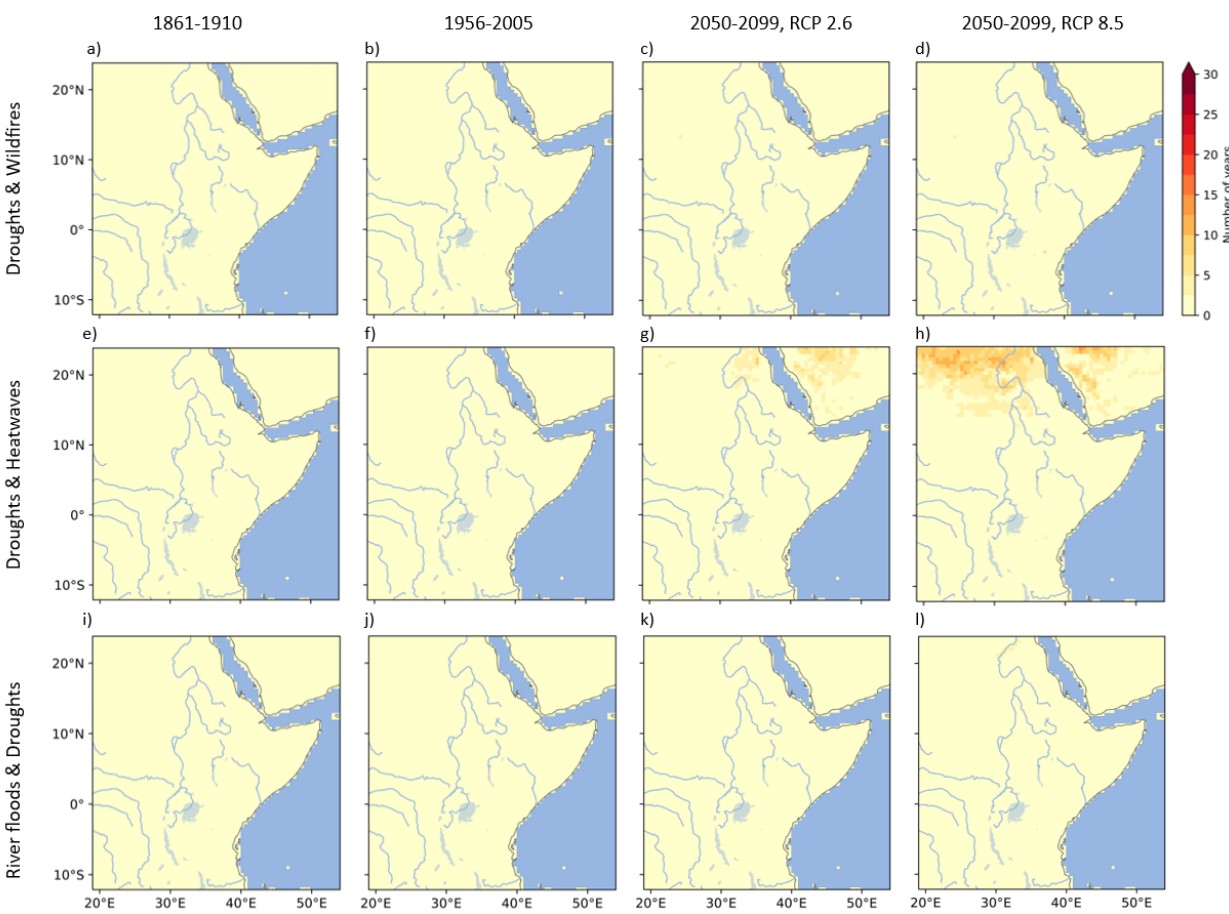

**Figure B1.** Average maximum number of consecutive years with joint occurrence of droughts & wildfires [a-d], droughts & heatwaves [e-h], and river floods & droughts [i-l], across 50-year time periods (columns) representing the early-industrial period (1861-1910), the present day (1956-2005), and the end of century (2050-2099) under RCP2.6 and RCP8.5. The average maximum number of consecutive years with joint occurrence of extremes represents the multi-model ensemble mean across all available combinations of extreme event simulations in the dataset driven by the same GCM

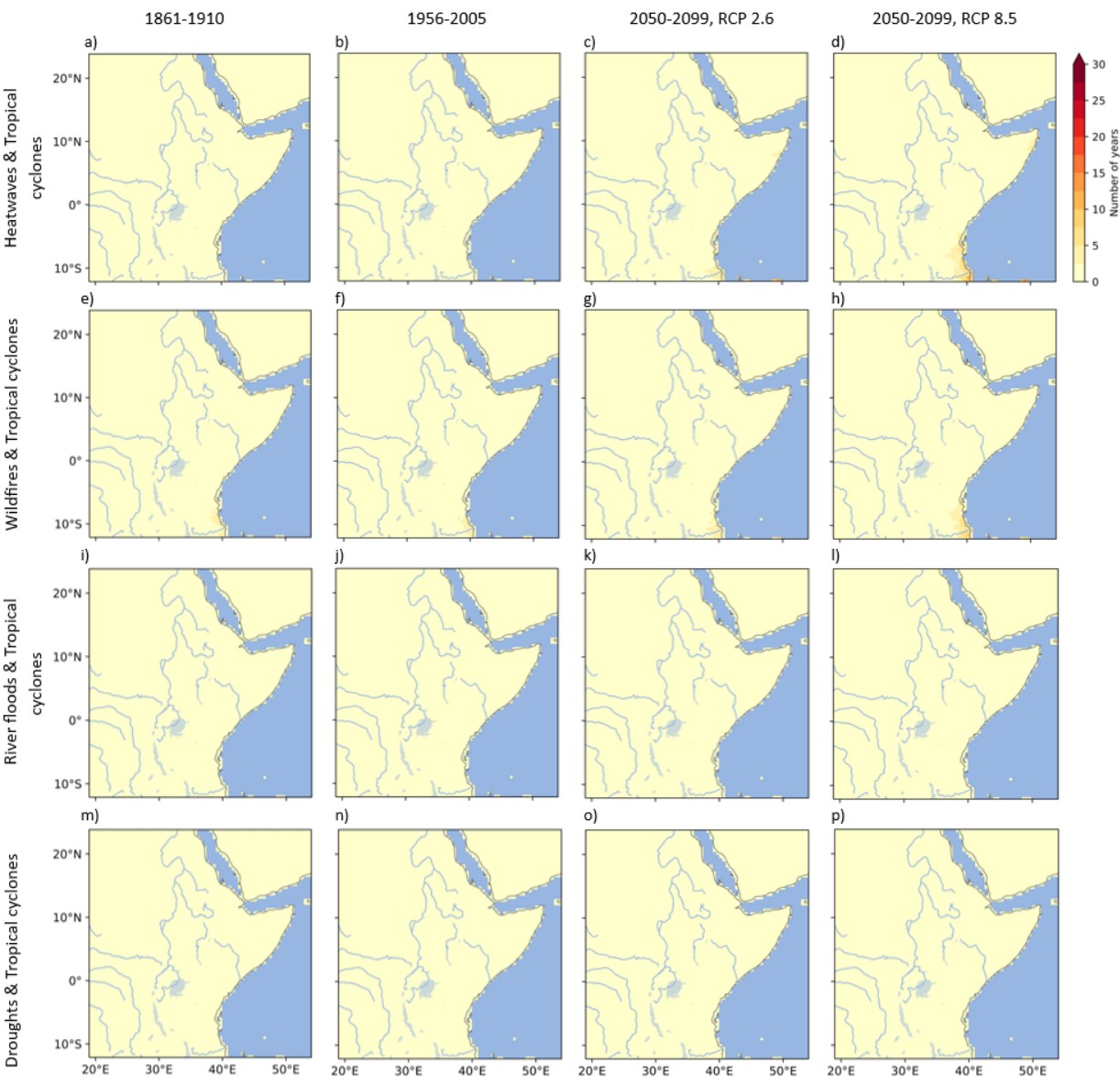

**Figure B2.** Average maximum number of consecutive years with joint occurrence of heatwaves & tropical cyclones [a-d], wildfires & tropical cyclones [e-h], river floods & tropical cyclones [i-l], and droughts & tropical cyclones [m-p], across 50-year time periods (columns) representing the early-industrial period (1861-1910), the present day (1956-2005), and the end of century (2050-2099) under RCP2.6 and RCP8.5. The average maximum number of consecutive years with joint occurrence of extremes represents the multi-model ensemble mean across all available combinations of extreme event simulations in the dataset driven by the same GCM

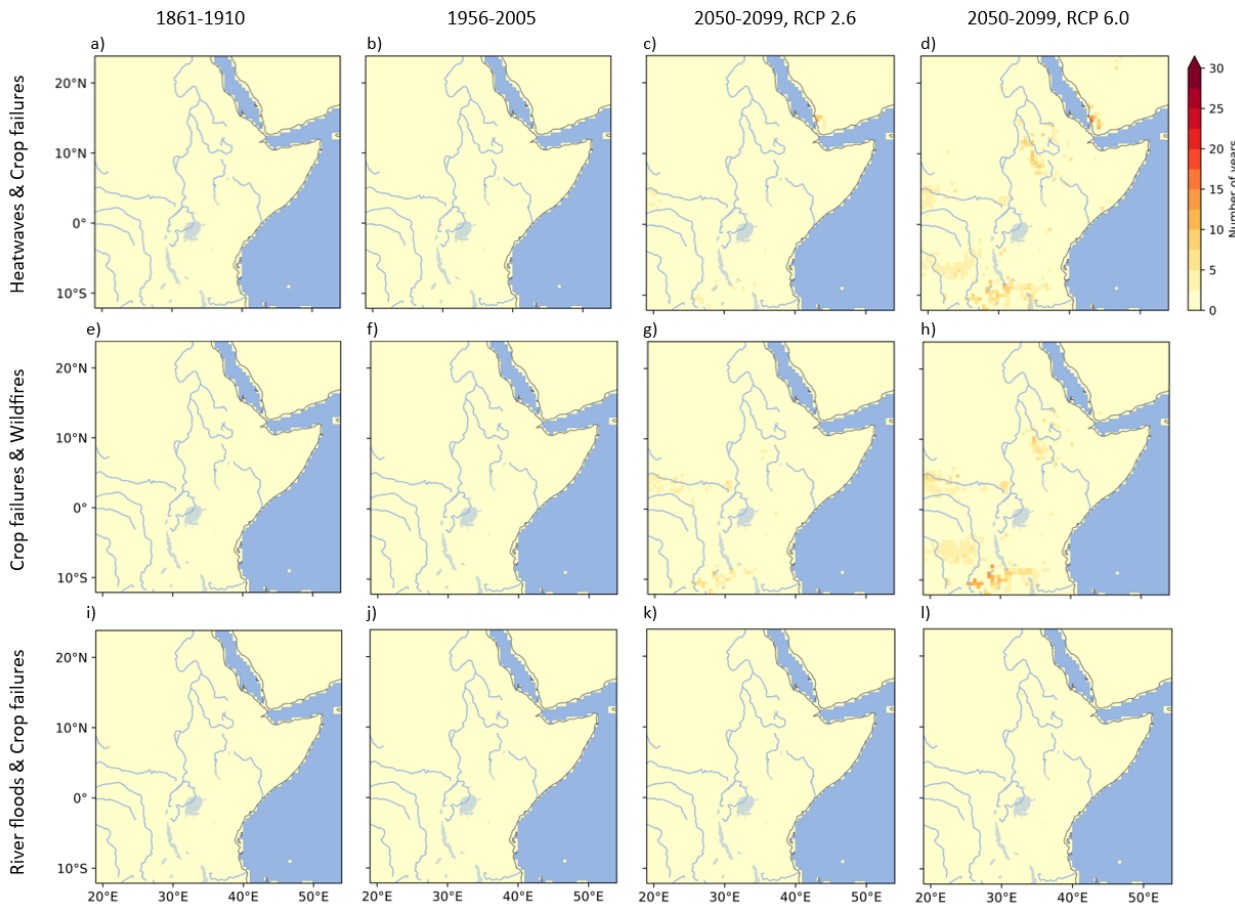

**Figure B3.** Average maximum number of consecutive years with joint occurrence of heatwaves & crop failures [a-d], crop failures & wildfires [e-h], and river floods & crop failures [i-l], across 50-year time periods (columns) representing the early-industrial period (1861-1910), the present day (1956-2005), and the end of century (2050-2099) under RCP2.6 and RCP6.0. The average maximum number of consecutive years with joint occurrence of extremes represents the multi-model ensemble mean across all available combinations of extreme event simulations in the dataset driven by the same GCM

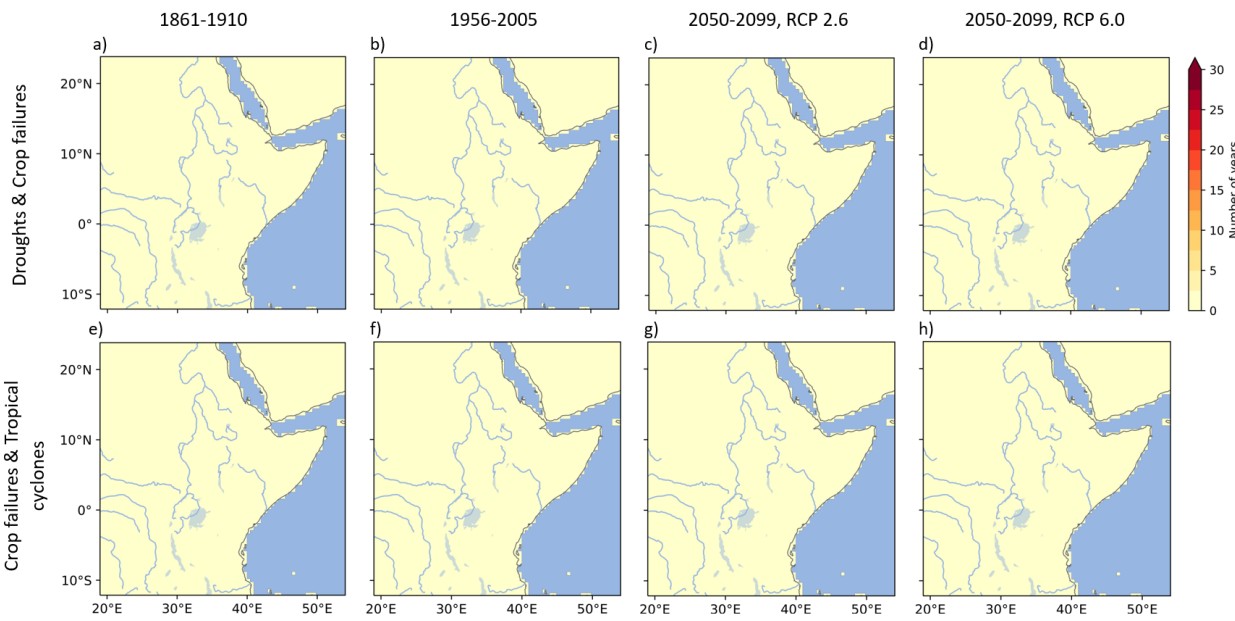

**Figure B4.** Average maximum number of consecutive years with joint occurrence of droughts & crop failures [a-d], and crop failures & tropical cyclones [e-h], across 50-year time periods (columns) representing the early-industrial period (1861-1910), the present day (1956-2005), and the end of century (2050-2099) under RCP2.6 and RCP6.0. The average maximum number of consecutive years with joint occurrence of extremes represents the multi-model ensemble mean across all available combinations of extreme event simulations in the dataset driven by the same GCM

**Appendix C: Bivariate distribution plots**

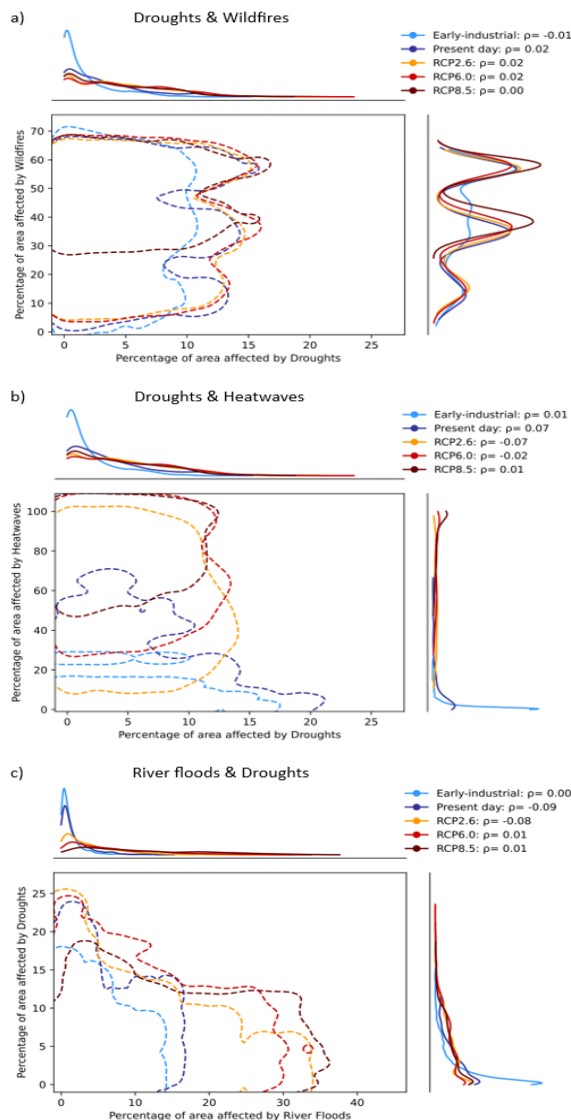

**Figure C1.** Bivariate distribution of: (a) droughts & wildfires, (b) droughts & heatwaves, and (c) river floods & droughts, across 50-year time periods representing the early-industrial period (1861-1910), the present day (1956-2005), and the end of century (2050-2099) under RCP2.6, 6.0, and 8.5. The marginal distributions of each extreme event (per scenario), based on the KDE method (Węglarczyk, 2018), are shown along the top and right axes of the plots. The contours (dotted lines) illustrate smooth estimates of the underlying distribution of co-occurring extremes. Here, the 68[th] percentile contour, which envelopes data within one standard deviation to either side of the mean, per scenario is used to show a generalized view of the distribution of the percentage of area affected by co-occurring extremes per year during the respective scenarios.

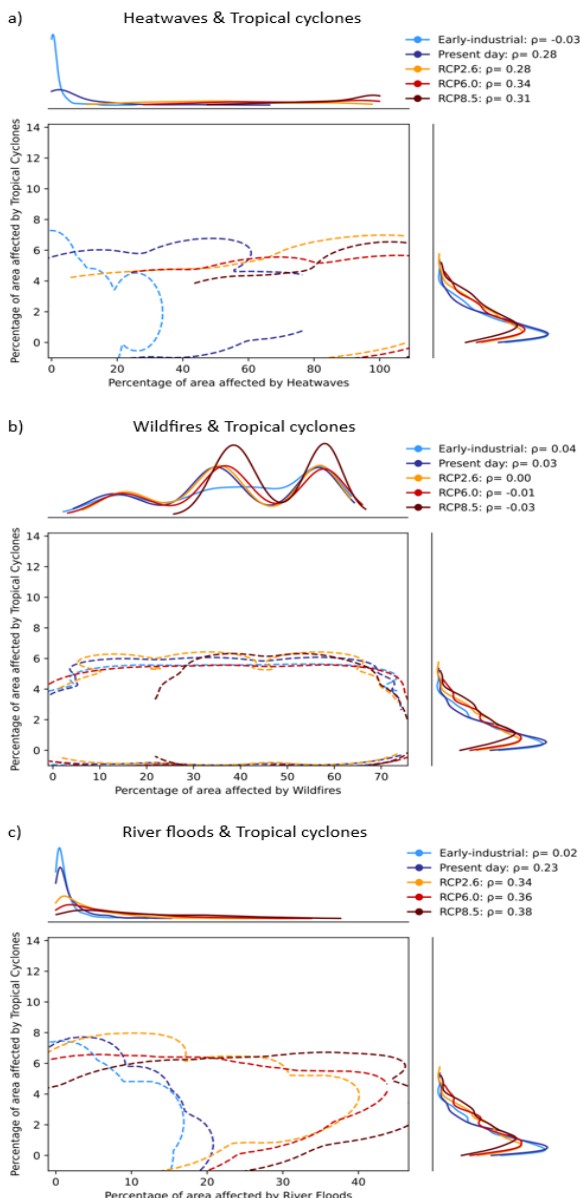

**Figure C2.** Bivariate distribution of: (a) heatwaves & tropical cyclones, (b) wildfires & tropical cyclones, and (c) river floods & tropical cyclones, across 50-year time periods representing the early-industrial period (1861-1910), the present day (1956-2005), and the end of century (2050-2099) under RCP2.6, 6.0, and 8.5. The marginal distributions of each extreme event (per scenario), based on the KDE method (Węglarczyk, 2018), are shown along the top and right axes of the plots. The contours (dotted lines) illustrate smooth estimates of the underlying distribution of co-occurring extremes. Here, the 68[th] percentile contour, which envelopes data within one standard deviation to either side of the mean, per scenario is used to show a generalized view of the distribution of the percentage of area affected by co-occurring extremes per year during the respective scenarios.

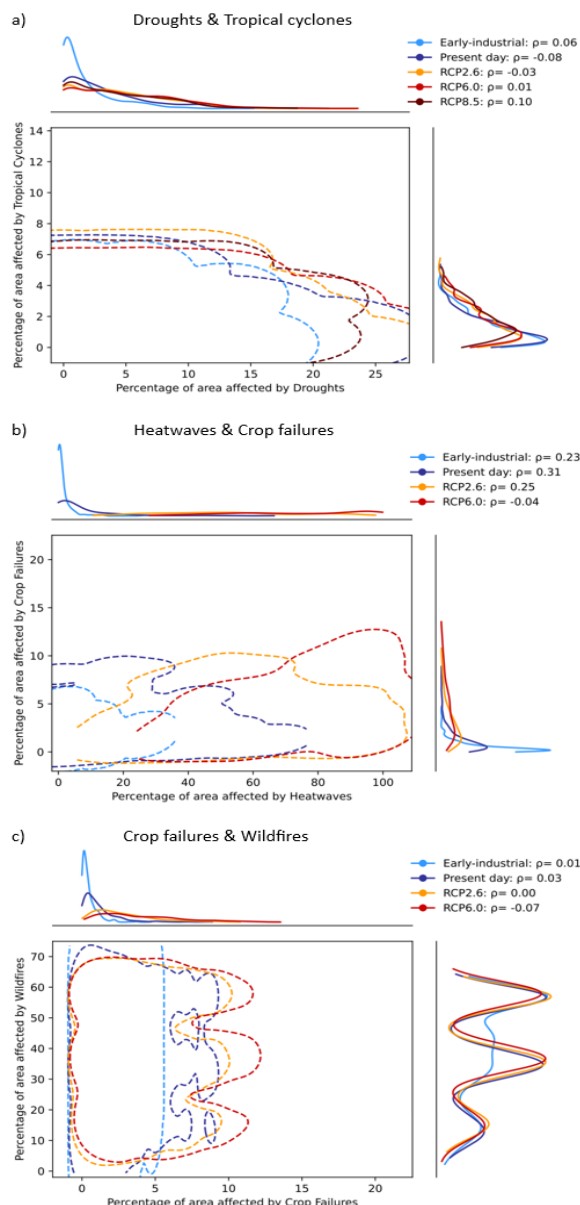

**Figure C3.** Bivariate distribution of: (a) droughts & tropical cyclones, (b) heatwaves & crop failures, and (c) crop failures & wildfires, across 50-year time periods representing the early-industrial period (1861-1910), the present day (1956-2005), and the end of century (2050-2099) under RCP2.6, 6.0, and 8.5. The marginal distributions of each extreme event (per scenario), based on the KDE method (Węglarczyk, 2018), are shown along the top and right axes of the plots. The contours (dotted lines) illustrate smooth estimates of the underlying distribution of co-occurring extremes. Here, the 68[th] percentile contour, which envelopes data within one standard deviation to either side of the mean, per scenario is used to show a generalized view of the distribution of the percentage of area affected by co-occurring extremes per year during the respective scenarios.

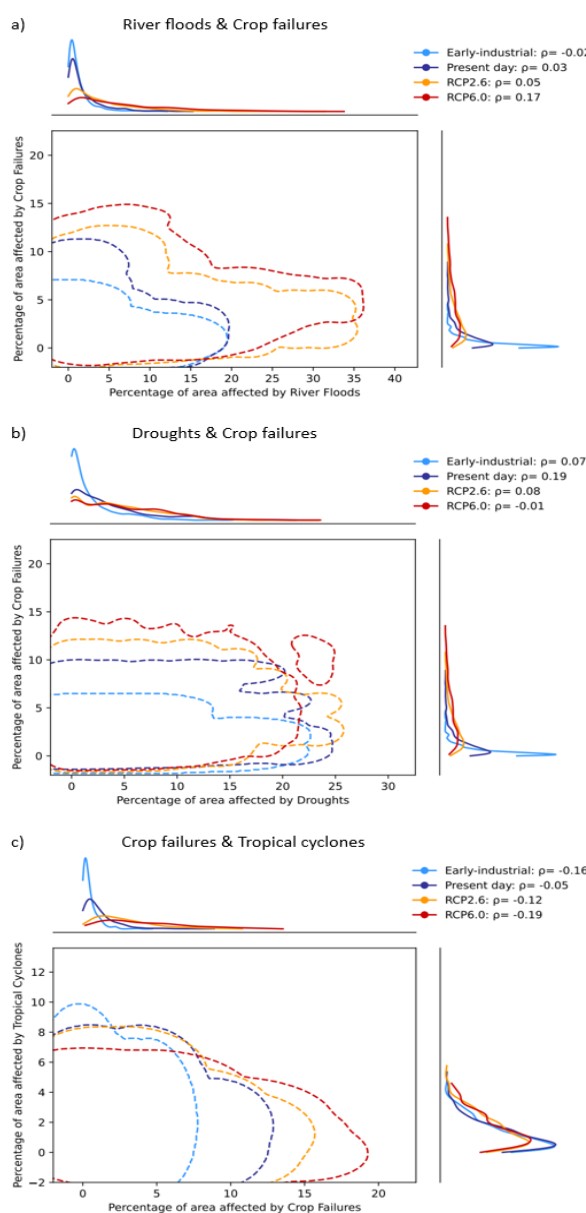

**Figure C4.** Bivariate distribution of: (a) river floods & crop failures, (b) droughts & crop failures, and (c) crop failures & tropical cyclones, across 50-year time periods representing the early-industrial period (1861-1910), the present day (1956-2005), and the end of century (2050-2099) under RCP2.6 and 6.0. The marginal distributions of each extreme event (per scenario), based on the KDE method (Węglarczyk, 2018), are shown along the top and right axes of the plots. The contours (dotted lines) illustrate smooth estimates of the underlying distribution of co-occurring extremes. Here, the 68[th] percentile contour, which envelopes data within one standard deviation to either side of the mean, per scenario is used to show a generalized view of the distribution of the percentage of area affected by co-occurring extremes per year during the respective scenarios.

## Appendix D: Determinants of changes in co-occurring extremes

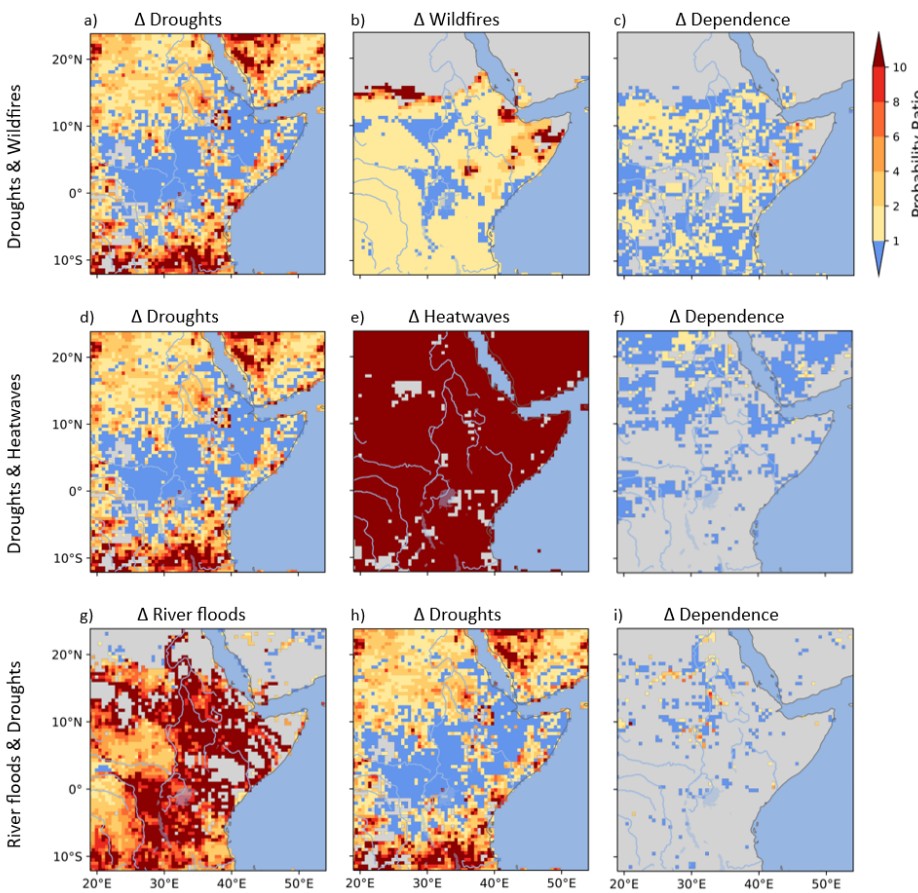

**Figure D1.** Determinants of change in co-occurring extreme event occurrence. Contributing PRs to the change in probability of joint occurrence of: droughts & wildfires [a-c], droughts & heatwaves [d-f], and river floods & droughts [g-i], whereby we illustrate the PR assuming only change in one of the extremes per pair (first and second column) and the PR assuming change only in the dependence of two co-occurring extremes (third column). The resulting PRs compare the end-of-century conditions under RCP8.5 to the early-industrial period conditions, whereby $PR \geq 1$ represents more likely occurrence of the extremes and $PR < 1$ represents less likely occurrence. The grey shaded areas represent areas that did not experience at least one of the extreme events per pair during the early-industrial period.

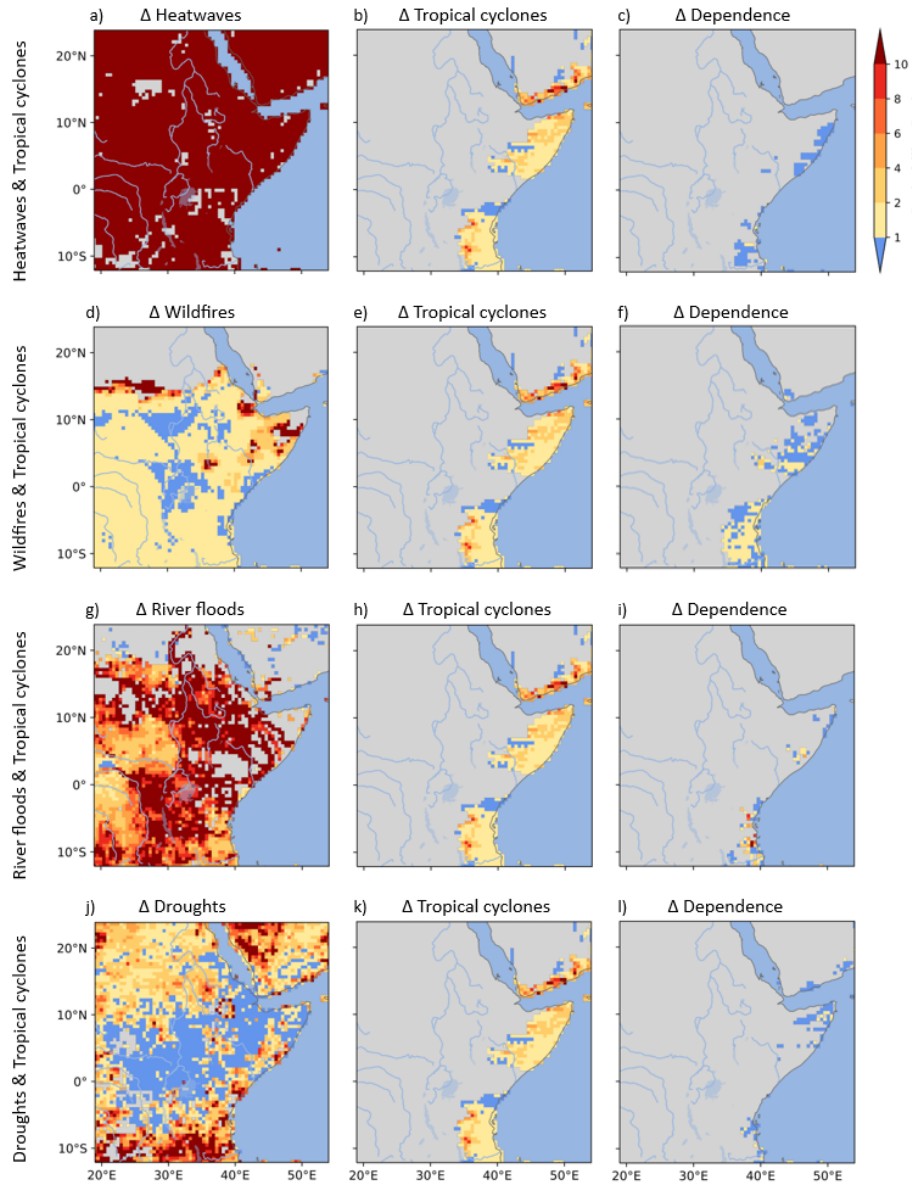

**Figure D2.** Determinants of change in co-occurring extreme event occurrence. Contributing PRs to the change in probability of joint occurrence of: heatwaves & tropical cyclones [a-c], wildfires & tropical cyclones [d-f], river floods & tropical cyclones [g-i], and droughts & tropical cyclones [j-l], whereby we illustrate the PR assuming only change in one of the extremes per pair (first and second column) and the PR assuming change only in the dependence of two co-occurring extremes (third column). The resulting PRs compare the end-of-century conditions under RCP8.5 to the early-industrial period conditions, whereby $PR \geq 1$ represents more likely occurrence of the extremes and $PR < 1$ represents less likely occurrence. The grey shaded areas represent areas that did not experience at least one of the extreme events per pair during the early-industrial period.

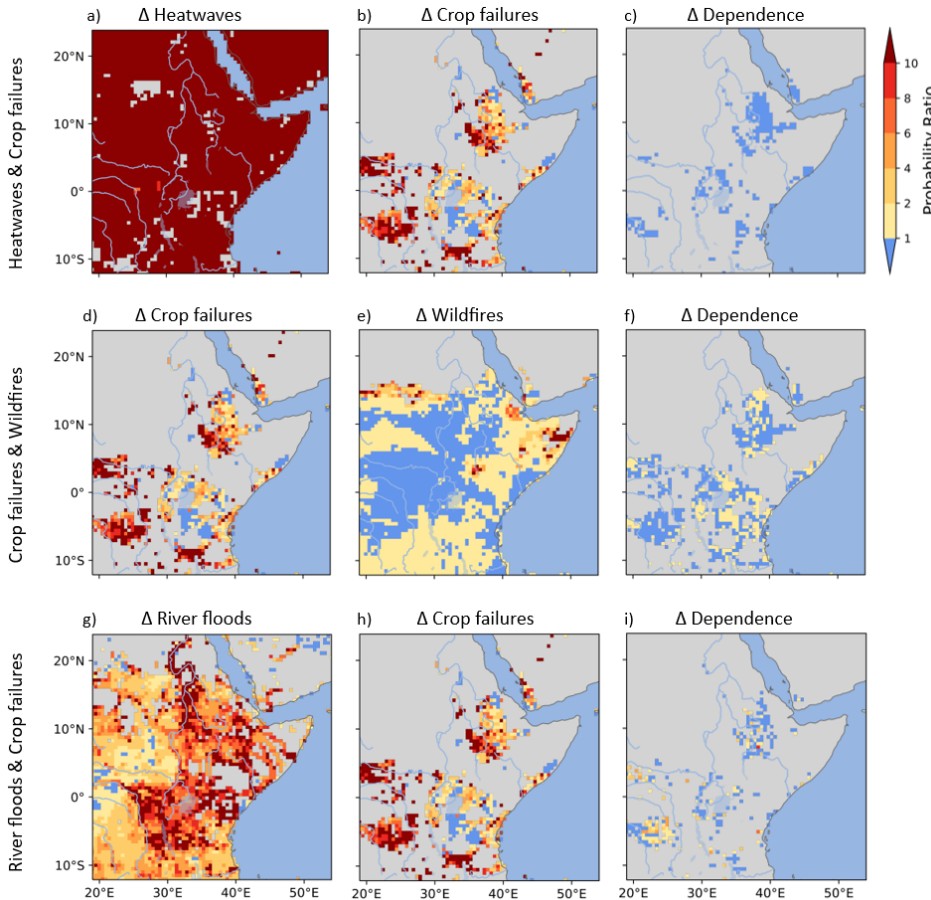

**Figure D3.** Determinants of change in co-occurring extreme event occurrence. Contributing PRs to the change in probability of joint occurrence of: heatwaves & crop failures [a-c], crop failures & wildfires [d-f], and river floods & crop failures [g-i], whereby we illustrate the PR assuming only change in one of the extremes per pair (first and second column) and the PR assuming change only in the dependence of two co-occurring extremes (third column). The resulting PRs compare the end-of-century conditions under RCP6.0 to the early-industrial period conditions, whereby $PR \geq 1$ represents more likely occurrence of the extremes and $PR < 1$ represents less likely occurrence. The grey shaded areas represent areas that did not experience at least one of the extreme events per pair during the early-industrial period.

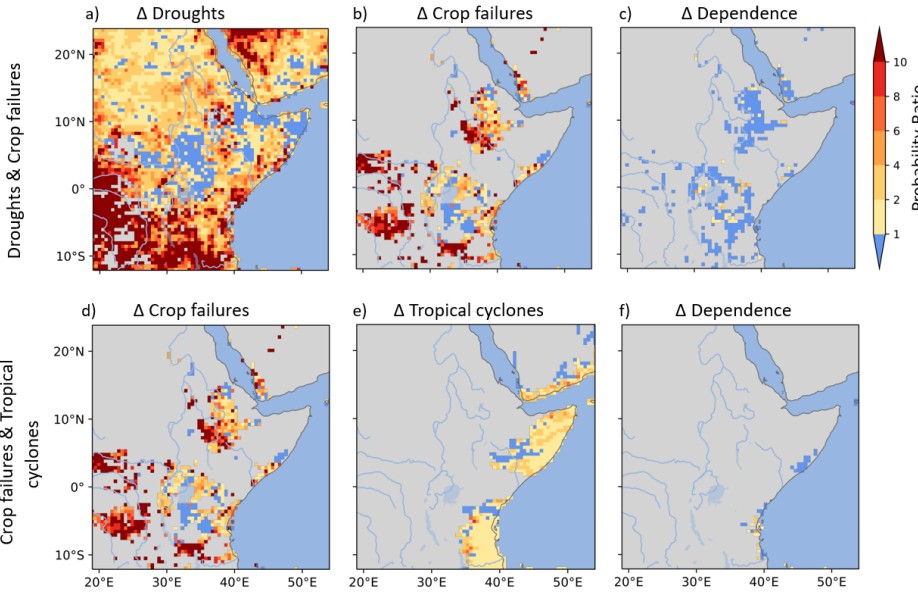

**Figure D4.** Determinants of change in co-occurring extreme event occurrence. Contributing PRs to the change in probability of joint occurrence of: droughts & crop failures [a-c] and, crop failures & tropical cyclones [d-f], whereby we illustrate the PR assuming only change in one of the extremes per pair (first and second column) and the PR assuming change only in the dependence of two co-occurring extremes (third column). The resulting PRs compare the end-of-century conditions under RCP6.0 to the early-industrial period conditions, whereby $PR \geq 1$ represents more likely occurrence of the extremes and $PR < 1$ represents less likely occurrence. The grey shaded areas represent areas that did not experience at least one of the extreme events per pair during the early-industrial period.

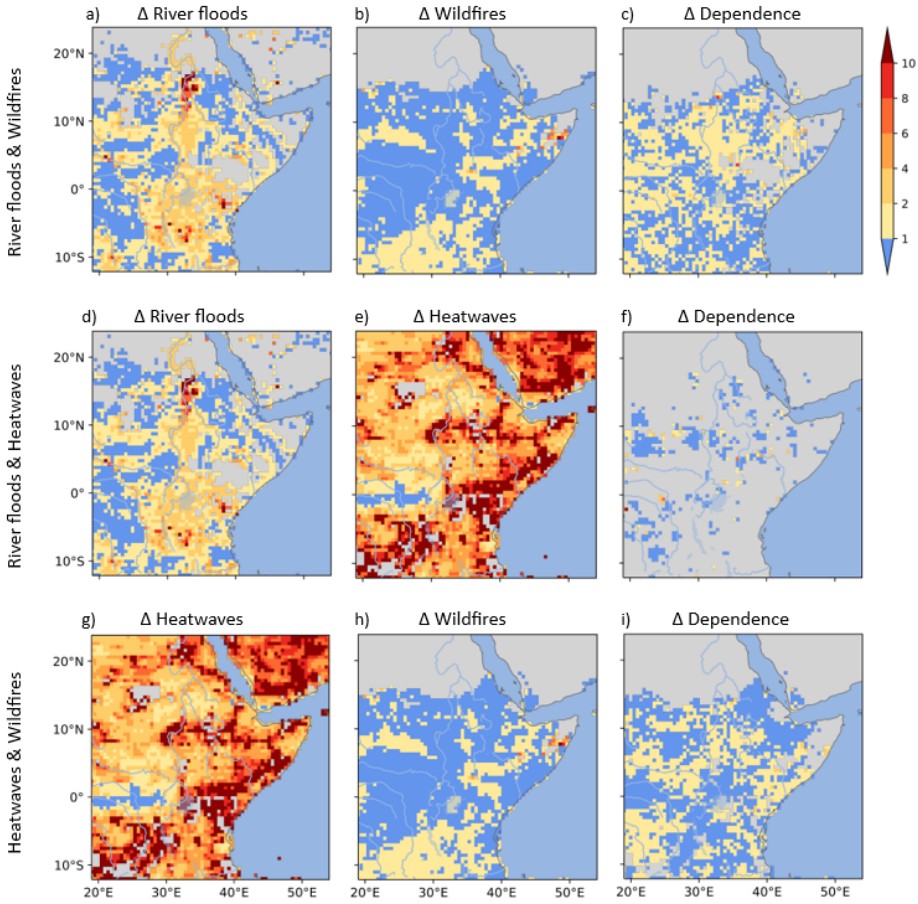

**Figure D5.** Determinants of change in co-occurring extreme event occurrence. Contributing PRs to the change in probability of joint occurrence of: river floods & wildfires [a-c], river floods & heatwaves [d-f], and heatwaves & wildfires [g-i], whereby we illustrate the PR assuming only change in one of the extremes per pair (first and second column) and the PR assuming change only in the dependence of two co-occurring extremes (third column). The resulting PRs compare present-day climate to the early-industrial period conditions, whereby $PR \geq 1$ represents more likely occurrence of the extremes and $PR < 1$ represents less likely occurrence. The grey shaded areas represent areas that did not experience at least one of the extreme events per pair during the early-industrial period.

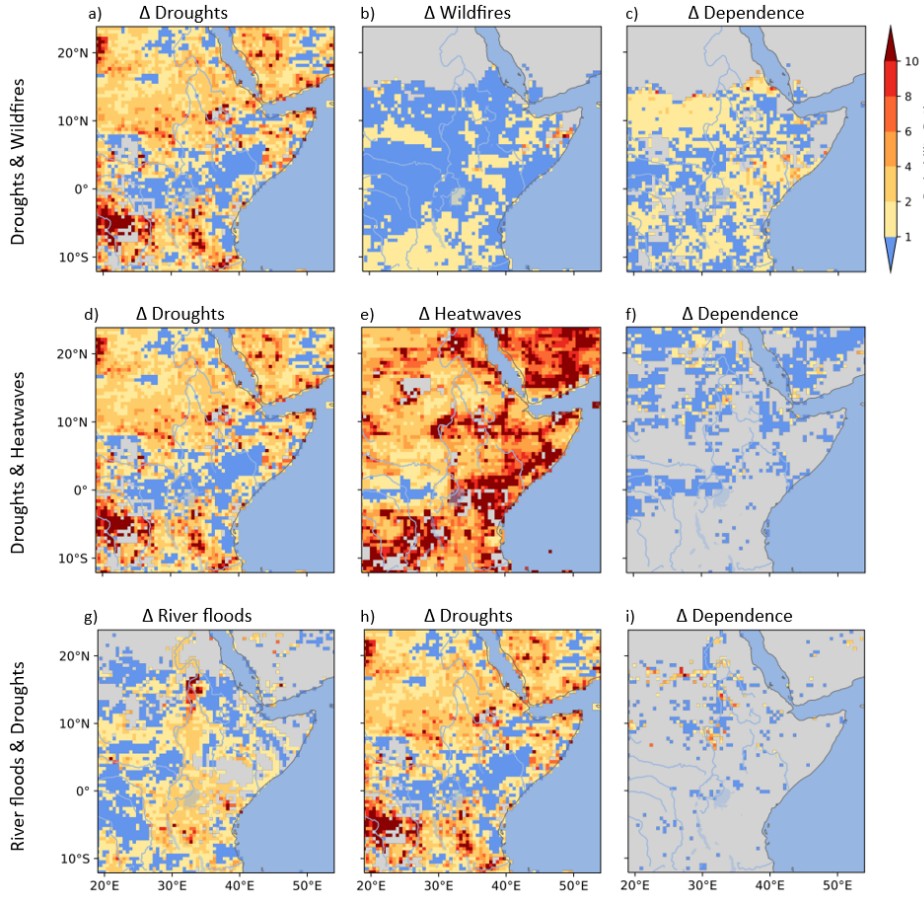

**Figure D6.** Determinants of change in co-occurring extreme event occurrence. Contributing PRs to the change in probability of joint occurrence of: droughts & wildfires [a-c], droughts & heatwaves [d-f], and river floods & droughts [g-i], whereby we illustrate the PR assuming only change in one of the extremes per pair (first and second column) and the PR assuming change only in the dependence of two co-occurring extremes (third column). The resulting PRs compare present-day climate to the early-industrial period conditions, whereby $PR \geq 1$ represents more likely occurrence of the extremes and $PR < 1$ represents less likely occurrence. The grey shaded areas represent areas that did not experience at least one of the extreme events per pair during the early-industrial period.

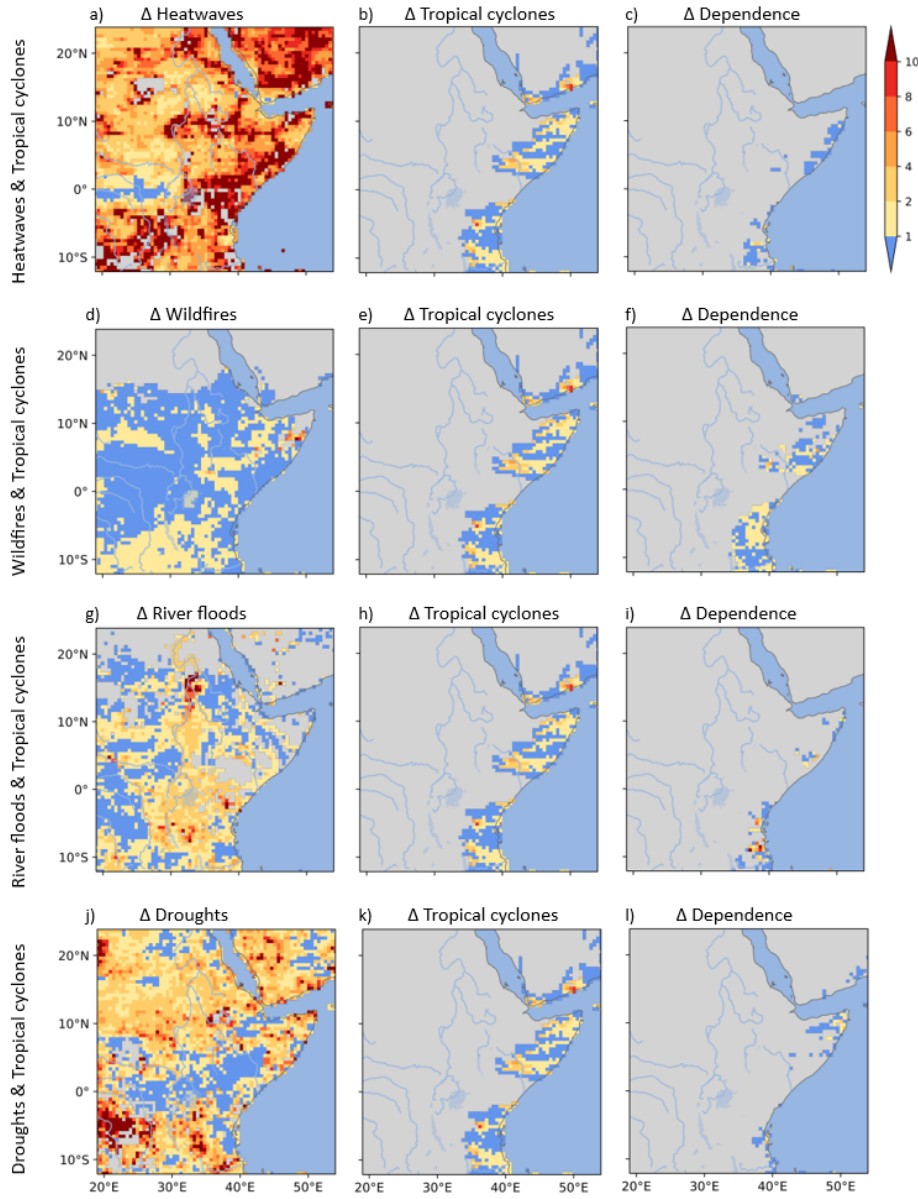

**Figure D7.** Determinants of change in co-occurring extreme event occurrence. Contributing PRs to the change in probability of joint occurrence of: heatwaves & tropical cyclones [a-c], wildfires & tropical cyclones [d-f], river floods & tropical cyclones [g-i], and droughts & tropical cyclones [j-l], whereby we illustrate the PR assuming only change in one of the extremes per pair (first and second column) and the PR assuming change only in the dependence of two co-occurring extremes (third column). The resulting PRs compare present-day climate to the early-industrial period conditions, whereby $PR \geq 1$ represents more likely occurrence of the extremes and $PR < 1$ represents less likely occurrence. The grey shaded areas represent areas that did not experience at least one of the extreme events per pair during the early-industrial period.

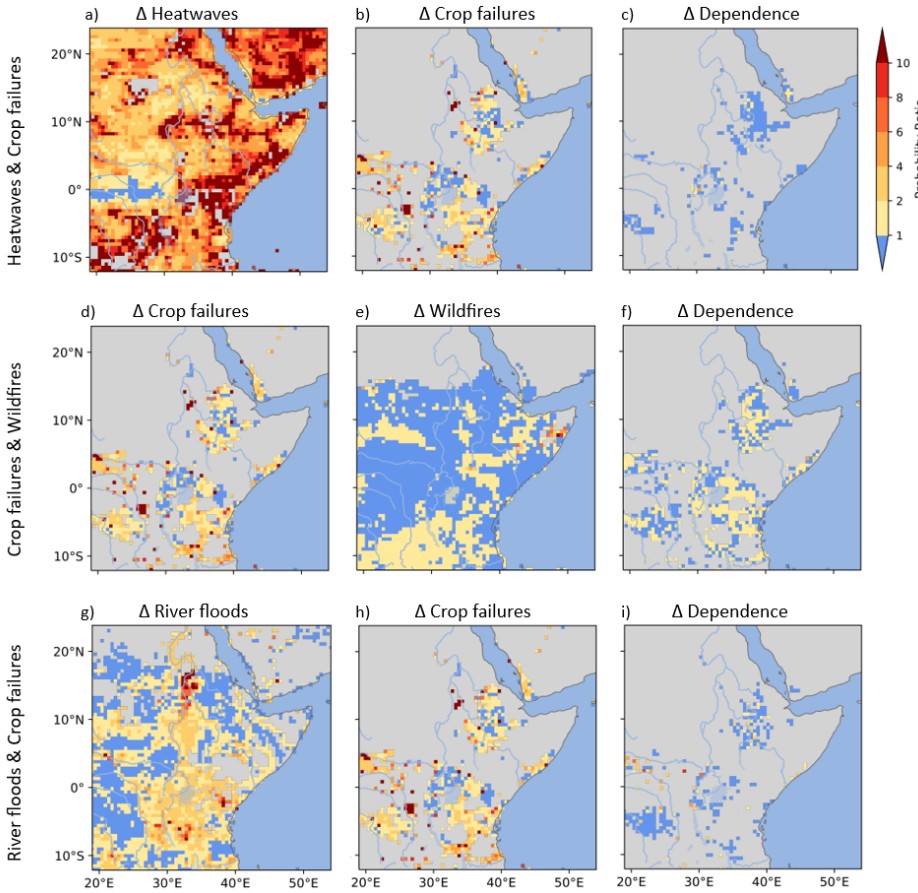

**Figure D8.** Determinants of change in co-occurring extreme event occurrence. Contributing PRs to the change in probability of joint occurrence of: heatwaves & crop failures [a-c], crop failures & wildfires [d-f], and river floods & crop failures [g-i], whereby we illustrate the PR assuming only change in one of the extremes per pair (first and second column) and the PR assuming change only in the dependence of two co-occurring extremes (third column). The resulting PRs compare present-day climate to the early-industrial period conditions, whereby $PR \geq 1$ represents more likely occurrence of the extremes and $PR < 1$ represents less likely occurrence. The grey shaded areas represent areas that did not experience at least one of the extreme events per pair during the early-industrial period.

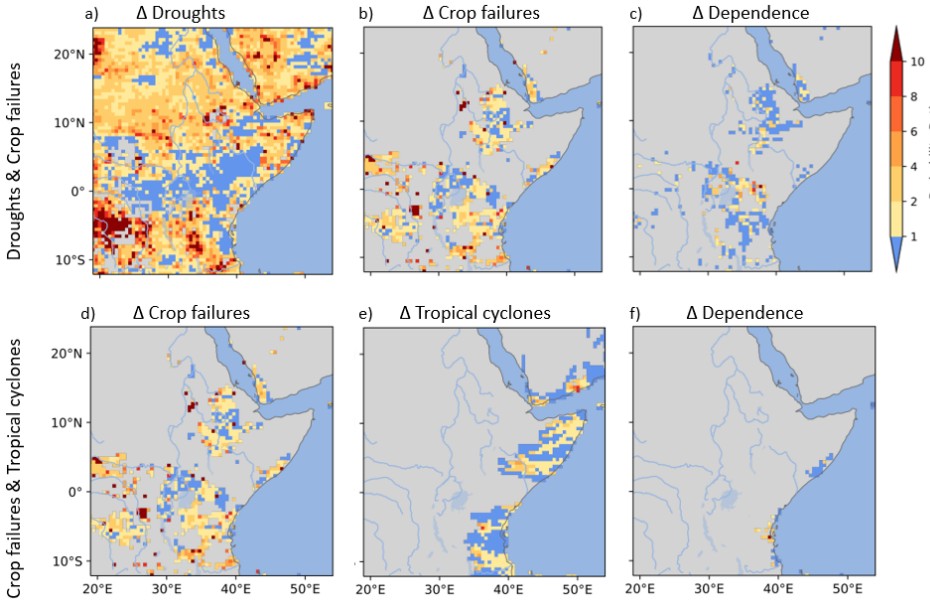

**Figure D9.** Determinants of change in co-occurring extreme event occurrence. Contributing PRs to the change in probability of joint occurrence of: droughts & crop failures [a-c] and, crop failures & tropical cyclones [d-f], whereby we illustrate the PR assuming only change in one of the extremes per pair (first and second column) and the PR assuming change only in the dependence of two co-occurring extremes (third column). The resulting PRs compare present-day climate to the early-industrial period conditions, whereby $PR \geq 1$ represents more likely occurrence of the extremes and $PR < 1$ represents less likely occurrence. The grey shaded areas represent areas that did not experience at least one of the extreme events per pair during the early-industrial period.

*Author contributions.* W.T., A.A.J.D, and D.M. designed the study. D.M. developed the scripts and conducted the data analysis with help from A.A., J.D, W.T., E.B., G.M., and J.Z.. All authors provided feedback during the data analysis stage and throughout the paper writing.

*Competing interests.* Some authors are members of the editorial board of Earth System Dynamics. The peer-review process was guided by an independent editor, and the authors have also no other competing interests to declare.

*Disclaimer.* TEXT

*Acknowledgements.* We would like to acknowledge the Inter-Sectoral Impact Model Intercomparison Project (ISIMIP; https://www.isimip.org/) for their part in mobilizing several cross-sectoral modelling groups whose climate-impact simulations provide a cohesive and comprehensive depiction of the world under different climate change scenarios, which are key in this research. Derrick Muheki is a research fellow at the Research Foundation Flanders (11M8823N). Special thanks goes to the authors from Lange et al. (2020) whose postprocessed ISIMIP2b dataset was key in the analysis of the co-occurring climate extremes in East Africa. The resources and services used in this work were provided by the VSC (Flemish Supercomputer Center), funded by the Research Foundation - Flanders (FWO) and the Flemish Government. This project has received funding from the European Union's Horizon 2020 and Horizon Europe research and innovation programmes under grant agreement nos. 101003469 and 101112727.

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
