# Peer review of "The perfect storm? Co-occurring climate extremes in East Africa"

_EGUsphere, 2023_

## Author Response (AR1)

We thank the reviewers for their time and valuable suggestions to improve the manuscript. Here below, we respond to each individual comment and propose modifications to the manuscript to accommodate the concerns raised. The following convention is used in this document to illustrate the text modifications in the original manuscript: modified text. Additionally, we mention new changes that we implemented in the revised manuscript.

**1  Anonymous Referee 1**

**1.1  Anonymous Referee 1 Comment**

> The paper aims at understanding concurrent climate extremes over the eastern parts of Africa using three different emission scenarios. The results show that, concurrent climate extremes are likely to increase with high magnitude over the Nile and Congo basin that are currently the wet regions over East Africa. Heat waves are wildfires are likely to dominate the region by the end of the 21st century as projected by all the emission scenarios.
>
> The paper is coherent, the methods deployed are relevant and the paper conceptualization is well thought of.
>
> Hence, I recommend the paper to be accepted for publication in EGU with minor correction.
>
> **Minor Correction.**
>
> The authors should enhance labelling of Lat and Lon in all the spatial maps since they are not currently clear.

**Response:** We thank you the reviewer for their feedback. We have revised the labelling of the Latitudes and Longitudes in all the spatial maps (Fig 2, 3, 5, A1-A4, B1-B4, and D1-D4) by changing the font color of the labels from grey to black and increasing their font size, to ensure that they are clear.

**2  Anonymous Referee 2**

This paper analysed the frequency and spatial extent of 15 types of concurrent extreme events in East Africa, highlighted the concurrent extremes will become the norm in the future. However. It remains some issues to be discussed before it is considered for publication.

**2.1  Anonymous Referee 2 Comment 1**

> The pair of concurrent extreme events defined in this paper represent two extreme events occurred within the same location in the same year, no matter if it occurred once or several times. The physics meaning of this definition is unclear. Usually the concurrent extreme events are defined

based on daily data, i.e., a pair of extreme events occurring on the same day. The definition in this paper seems to be too crude to obtain physically-meaningful knowledge.

**Response:** We thank the reviewer for this comment. There are three distinct motivations for our methodological choice to consider annual time steps in this study. The first concerns data availability: the available dataset from ISIMIP 2b only provides annual occurrences of the six categories of climate extreme events (as mentioned in one of the caveats in lines 79-80 of the manuscript). Unfortunately, no multi-hazard future data is available at daily resolution. Due to this limitation of the dataset, we define concurrent events as two events occurring in the same location, during the same time step (here being the same year). A second reason is that some of the climate extremes we consider play out over longer time scales: for example, droughts may last several months to even years, wildfires may rage an entire summer season, and crop failures may result from extreme conditions during the entire growing season. A third motivation is that the impacts of compound extremes may be larger than those for individual events even in the case where the concurrence is not on a daily timescale. These are sometimes termed temporally compounding extremes (e.g. Zscheischler et al., 2020), although for simplicity we opted not to make this terminological distinction in our study. For example, vegetation impacts of drought events can be aggravated by droughts in consecutive growing seasons (e.g. Bastos et al., 2021). Similarly, societal vulnerability to floods is modulated by the occurrence of successive flood episodes (Chacowry et al., 2018). We therefore argue that there is some relevance to considering concurrent extremes on yearly timescales.

We nonetheless agree that being able to discriminate the timescale of concurrence of the extreme events we study would allow a more nuanced analysis, as illustrated in lines 23-30. Similarly, we agree with the values of knowing whether a given extreme has occurred once or several times in a given year. We updated the original manuscript to clarify these messages throughout the revised text in Sect. 2 lines 78-87, as follows:

> The dataset we use comes with a number of caveats. A minor caveat is that it does not contain crop failure projections under RCP8.5. More importantly, the data represents the occurrence of an extreme event category as a single event within a grid cell per year, no matter if it occurred once or several times within the same location in the same year. Finally, an extreme event such as a wildfire, river flood or tropical cyclone can only partly cover a given grid cell, whereas other extreme events (heatwaves, droughts and crop failures) are assigned by default to the entire grid cell. Thus, for the former three extremes, we consider that a grid cell is entirely affected when more than 0.5% of the 0.5°x0.5° grid cell area is simulated to be affected by the extreme event. Whilst these are limitations of the dataset, we have three distinct motivations to use it throughout our analysis: (i) the dataset is amongst the most detailed and complete of its kind, and provides information on the

occurrence of extreme events within the study region over a very long time period (from 1861 until 2099); (ii) some of the climate extremes we consider play out over longer time scales, for example droughts may last several months to even years, wildfires may rage an entire summer season, and crop failures may result from extreme conditions during the entire growing season; (iii) the impacts of compound extremes may be larger than those for individual events even in the case where the concurrence is not on a daily timescale. These are sometimes termed temporally compounding extremes (e.g., Zscheischler et al. 2020b). For example, vegetation impacts of drought events can be aggravated by droughts in consecutive growing seasons (e.g., Bastos et al. 2021). Similarly, societal vulnerability to floods is modulated by the occurrence of successive flood episodes (Chacowry et al., 2018). We therefore use it as the backbone for this study.

**2.2 Anonymous Referee 2 Comment 2**

> From fig.1, among the 15 pairs of extremes, all the concurrent extremes including heatwaves show the strongest increase. It this related to the apparent global warming trend? And all the future years will become extreme years of heatwaves?

**Response**: Yes, the increases shown in Fig.1 are due to the global warming trend. However, while our analysis shows that frequency of heatwaves will increase drastically, not all future years are projected as extreme heatwave years for every location (grid) in East Africa, even under the high warming scenarios (*Illustration 1*). As reported by many studies, the frequency and magnitude of heatwaves is projected to increase in many regions of the world as a result of global warming (e.g., Russo et al. 2014, 2015, Thiery et al. 2021, IPCC 2021). This is generally confirmed in our analysis over East Africa (Table 2), whereby the mean percentage area affected by heatwaves is projected to increase by the end of the century under all the three future climate scenarios, with larger increases in the warmer scenarios. Furthermore, the increase in probability and spatial extent of concurrent extremes including heatwaves (in the pairs) is projected to increase, with increases in heatwaves identified as a main driver of these events. This is illustrated in lines 290-297 of the revised manuscript. While we indeed find that the change in heatwave occurrence provides a strong contribution to the change in occurrence of concurrent extremes, this is far from our only result/conclusion. Indeed, we also highlight river floods and wildfires as a pair of extremes whose concurrence will increase sharply (Fig. 5), and we analyse other pairs of concurrent extremes not involving heatwaves (Appendix D). We have added the following text in the method (Sect. 3.3 Line 125-127) to clarify that the changes in the concurrent extreme occurrence are projected as a result of global warming.:

Considering that the processed impact model simulations account only for climate-induced changes in the extremes (as defined by Lange et al. (2020)), and not for other changes such as land-use, here we only analyse the climate change-driven effects on concurrent extremes. At a given location, from a statistical perspective, the probability of concurrent extreme events can be affected by the effect of climate change on: (i) the probability of the individual extreme events and/or (ii) the dependence between the events (Bevacqua et al., 2020; Zscheischler et al., 2020b). To gain insights into the drivers of the changes, we compute the change in the probability of concurrent extreme events when assuming: (i) changes in the probability of extreme events in one variable only; and (ii) changes in the coupling between the variables only (Bevacqua et al., 2020).

[Figure]

**Illustration 1:** Average percentage of years projected with occurrence of heatwaves by the end of the century (2050-2099) under RCP2.6 [a], RCP6.0 [b] and RCP8.5 [c] scenarios. The average percentage of years with occurrence of heatwaves represents the multi-model ensemble mean across all available GCMs (considering one heatwave definition: HWMId99).

**2.3    Anonymous Referee 2 Comment 3**

> The method used in this paper to search drivers of concurrent extremes seems to be too shallow. So, the results look very obvious, i.e., the global warming is the most important driver. The paper lacks an analysis of the dynamic process that cause to concurrent extreme events

**Response:** Considering that the six extreme events considered each have different meteorological and physical drivers, and that we utilise extreme event data from processed impact model simulations, diving into the meteorological and physical drivers of the concurrent extreme events presents near-insurmountable challenges. We nonetheless believe that our analysis investigating

the changes in the probability of occurrence of concurrent events under future climate scenarios in comparison to early-industrial conditions. This is because: (i) The models only account for climate-induced changes in the hazards and not changes due to land-use change (e.g., deforestation fires) (Lange et al., 2020). Therefore, we can only look at climate change/global warming as the driver of the changes in extreme event occurrence, and in turn the changes in concurrent extremes. (ii) We, furthermore, go deeper in our analysis by splitting the change in concurrent extreme event occurrence into: changes in only one variable per pair, and the changes in the (coupling) dependence between extreme events in a pair, as illustrated in Section 3.3. This provides a good basis to formulate some physical hypotheses on the drivers of the changes in concurrent extreme events as illustrated in Sect. 5.3.

In the revised manuscript, we improved the communication of the method that we use to identify the drivers of changes in the concurrent extreme occurrences by adding the following text in the methods section (Sect. 3.3, Line 125-127) to clarify this point:

> Considering that the processed impact model simulations account only for climate-induced changes in the extremes (as defined by Lange et al. (2020)), and not for other changes such as land-use, here we only analyse the climate change-driven effects on concurrent extremes. At a given location, from a statistical perspective, the probability of concurrent extreme events can be affected by the effect of climate change on: (i) the probability of the individual extreme events and/or (ii) the dependence between the events (Bevacqua et al., 2020; Zscheischler et al., 2020b). To gain insights into the drivers of the changes, we compute the change in the probability of concurrent extreme events when assuming: (i) changes in the probability of extreme events in one variable only; and (ii) changes in the coupling between the variables only (Bevacqua et al., 2020).

Furthermore, in the discussions (Sect 5.3, Line 290-304), we added the following text to illustrate possible dynamical causes of change in river floods.

According to Niang et al. (2014) & Seneviratne et al. (2021), the East African region is also projected to experience increased intense precipitation by the end of the century (with high confidence) under RCP8.5 scenario. This could be linked to projected changes in large-scale modes of variability, such as the Indian Ocean Dipole and the El Niño-Southern Oscillation, which influence precipitation across East Africa and are already showing change under present-day conditions relative to the pre-industrial period (medium confidence, Seneviratne et al., 2021). In addition to large-scale teleconnections, projected changes in mesoscale circulation and local land-atmosphere feedbacks may further affect future precipitation patterns in the region (Souverijns et al., 2016).

**3  Other changes to the original manuscript**

**3.1  Table 1**

Here we edited the reference of the extreme event definitions (within the column headings) as follows:

> Definition in Lange et al. (2020)

**3.2  Text in Section 1**

We added one missing word in line 45 as shown below:

> This indicates the need for a detailed analysis of compound extremes in East Africa, that will allow for a better understanding of the possible dependencies between extreme events, their recurrence, and the effect of different future emission scenarios on their frequency. This is not only important for disaster risk management, but it is also key for climate change adaption planning by the government authorities in the region.

**3.3  Text in Section 3.3**

We edited the text describing the interpretation of Equation 7 as follows:

> Eq. 7 should be interpreted carefully when changes in $P(x)$ and/or $P(y)$ are large. In fact, as a caveat of 6 the fact that we deal with binary variables, by construction, when positive changes in $P(x)$ and/or $P(y)$ are large, the estimated future dependency tends to be small (i.e., $D(x,y)_{future} \simeq 1$) despite the continuous variables from which the binary variable X and Y possibly being coupled. This, in turn, affects the estimated P Rchange in D. However, we also note that under such potentially very large changes in $P(x)$ and/or $P(y)$, such changes control the actual change in the probability of concurrent extremes, and dependency changes become irrelevant (Bevacqua et al., 2022). In the case of very large negative changes in $P(x)$ and/or $P(y)$, the denominator in Eq. 7 would be very small, and thus it is not obvious to get a small future dependency. For a thorough assessment of the changes in the dependencies, continuous rather than binary variables X and Y (Bevacqua et al., 2020), as well as larger sample sizes (Bevacqua et al., 2023), would be required.

**3.4  Updated reference in Section 5.4**

We updated the previous 'In-press' reference on line 321 in the original manuscript with the reference for the recently published paper as follows:

> Further research into concurrent extremes in East Africa could also expand the methodology taken in this study to consider more than two extreme events occurring in the same location and year, and thus carry out a more complete multivariate 320 analysis. Additionally, we recommend the implementation of other metrics, such as propensity (Rosenbaum and Rubin,

1983) and co-occurrence ratio (Kornhuber and Messori, 2023), to further understand the occurrence of concurrent extremes in East Africa. Lastly, we recommend the application of our methods to other regions to illustrate how climate change may modulate concurrent extremes in different parts of the globe.

**4    Added References to the manuscript**

Bastos, A., Orth, R., Reichstein, M., Ciais, P., Viovy, N., Zaehle, S., Anthoni, P., Arneth, A., Gentine, P., Joetzjer, E., Lienert, S., Loughran, T., McGuire, P. C., O, S., Pongratz, J., and Sitch, S.: Vulnerability of European ecosystems to two compound dry and hot summers in 2018 and 2019, Earth Syst. Dynam., 12, 1015–1035, https://doi.org/10.5194/esd-12-1015-2021, 2021.

Chacowry, A., McEwen, L. J., & Lynch, K. (2018). Recovery and resilience of communities in flood risk zones in a small island developing state: A case study from a suburban settlement of Port Louis, Mauritius. International Journal of Disaster Risk Reduction, 28, 826-838, https://doi.org/https://doi.org/10.1016/j.ijdrr.2018.03.019, 2018.

Souverijns, N., Thiery, W., Demuzere M., and Van Lipzig N. P. M: Drivers of future changes in East African precipitation, Environ. Res. Lett, 11, https://doi.org/10.1088/1748-9326/11/11/114011, 2016

---

## Author Response (AR2)

**Response to Reviewer's comments**:

This paper investigates changes in the joint occurrence of two extreme climate events within the same year over East Africa from past to future climate conditions. The joint occurrence pairs of two events are selected from 6 individual events from the Inter-Sectoral Impact Model Intercomparison Project (ISIMIP) (Lange et al., 2020): river floods, droughts, heatwaves, crop failures, wildfires, and tropical cyclones. The authors also attempt to understand attributions of increases in the probability of concurrent events. Although the authors' analysis of concurrent events can potentially provide useful information for adaptation in a changing climate over East Africa, I think that the study design is problematic, and key results are already presented in Lange et al., 2020 while most other results are somewhat trivial. Therefore, substantial further work will be required to meet the standards of scientific publication. In case the editor judges this paper to be suitable for ESD or the authors consider resubmitting an improved manuscript, I provide my suggestions below.

We thank the reviewer for their time and valuable suggestions to improve the manuscript. Below, we respond to the individual comments and illustrate modifications to the manuscript to accommodate the concerns raised. We believe that the manuscript has benefited from these modifications. The following convention is used in this document to illustrate the text modifications in the original manuscript: modified text.

**Major comments**:

1. Drivers and Impacts

According to this paper's Abstract and Introduction, it appears that the goal of this study is to find past-to-future changes in compound (concurrent) events, and their interactions and dependence. Then, it would be logical to use a combination of meteorological and/or hydrological drivers happening simultaneously. (Think about the definition of a compound climate event as described even in this paper's Introduction.) However, the climate events used in this study are already "impacts" that originated in one or a combination of climate "drivers". Although some events such as heatwaves, droughts, or tropical cyclones can be considered as a driver in certain cases depending on the context of a study, events such as river floods, crop failures, and wildfires are impact-events from driver-events such as extreme precipitation, temperature, and winds. Therefore, it is strange and even wrong to say that "we determine the main drivers of changes in the occurrence of concurrent extremes" (Lines 59-60); the authors only compare combinations of impact events and do not study any physical mechanisms or drivers of such events. In my opinion, it is thus questionable of the validity of the author's definition of "D, Dependence" and its interpretation. For example, would the authors be able to provide a physical interpretation of D changes in Figure 5c? I'm not sure those values actually mean anything. This study itself is merely a display of a combination of two impact events happening within the "same year (!)", which provides an additional reason not to use the term "compound" or "concurrent". The title, Abstract, and Introduction are all misleading

**Response:**

We thank the reviewer for this valuable comment. There are three distinct motivations for our methodological choice to consider annual time steps in this study. The first concerns data availability: the available dataset from ISIMIP 2b only provides annual occurrences of the six categories of climate extreme events (as mentioned in one of the caveats in lines 79-80 of the manuscript). It is also important to note that this data is output from process-based impact models (as mentioned in lines 67-69). To the best of our knowledge, there is no publicly accessible future multi-hazard data source providing comparable information on e.g. daily or weekly timescales. Due

to this limitation of the dataset, we define concurrent events as two events occurring in the same location, during the same output data time step (here being the same year). Notwithstanding this data limitation, we still believe that defining compound events on a yearly basis does have some physical relevance, since some of the climate extremes we consider play out over long time scales. For example, droughts in our dataset are defined as the drop of soil water content below the 2.5$^{th}$ percentile of the distribution during pre-industrial times considering periods longer than 6 months (See Table 1), wildfires may rage for the better part of a summer season, and crop failures may result from extreme conditions during the entire growing season. A third motivation is that the impacts of compound extremes may be larger than those for individual events even where the concurrence is on a yearly scale. These are sometimes termed temporally compounding extremes (as per the compound event typology co-developed by part of the author team; Zscheischler et al., 2020), although for simplicity we opted not to make this terminological distinction in our study. For example, vegetation impacts of drought events can be aggravated by droughts in consecutive growing seasons (e.g. Bastos et al., 2021, Wu et al. 2022). Similarly, societal vulnerability to floods is modulated by the occurrence of successive flood episodes (Chacowry et al., 2018), and wildfires and hydrological extremes can also compound across seasons (Yu et al. 2023). We therefore argue that there is relevance to considering these extremes as concurrent even if based only on yearly timescales.

We nonetheless agree that being able to discriminate the timescale of concurrence of the extreme events we study would allow a more nuanced analysis, as illustrated in lines 26-32. Similarly, we agree with the value of knowing whether a given extreme has occurred once or several times in a given year. We updated the original manuscript to clarify these messages throughout the revised text in Sect. 2 lines 83-99, as follows:

> The dataset we use comes with several caveats. A minor caveat is that it does not contain crop failure projections under RCP8.5. More importantly, the data represents the occurrence of an extreme event category as a single event within a grid cell per year, no matter if it occurred once or several times within the same location in the same year. Finally, an extreme event such as a wildfire, river flood or tropical cyclone can only partly cover a given grid cell, whereas other extreme events (heatwaves, droughts and crop failures) are assigned by default to the entire grid cell. Thus, for the former three extremes, we consider that a grid cell is entirely affected when more than 0.5% of the 0.5°x0.5° grid cell area is simulated to be affected by the extreme event. Whilst these are limitations of the dataset, we have three distinct motivations to use it throughout our analysis: (i) the dataset is amongst the most detailed and complete of its kind, and provides information on the occurrence of extreme events within the study region over a very long time period (from 1861 until 2099); (ii) some of the climate extremes we consider play out over longer time scales, for example droughts may last several months to even years, wildfires may rage for several months, and crop failures may result from extreme conditions during the entire growing season; (iii) the impacts of compound extremes may be larger than those for individual events even in the case where the concurrence is not on a daily timescale. These are sometimes termed temporally compounding extremes (e.g., Zscheischler et al. 2020b). For example, impacts of drought events on vegetation can be aggravated by droughts in consecutive growing seasons (e.g., Bastos et al. 2021, Wu et al. 2022). Similarly, societal vulnerability to floods is modulated by the occurrence of successive flood episodes (Chacowry et al., 2018), and wildfires and hydrological extremes can also compound across seasons (Yu et al. 2023, Moody and Ebel, 2012; Larsen et al., 2009). We therefore use the yearly dataset as the backbone for this study.

Upon careful consideration, we have modified that the title, abstract and introduction to ensure that they are coherent and– accurately reflect the objectives and findings of our study. For the title we have adopted the term 'co-occurring extremes' instead of 'concurrent extremes'. This is because

'co-occurring' generally implies events happening within the same timeframe, but not necessarily on the exact same day as opposed to 'concurrent' that may imply events happening at the same time/simultaneously. Additionally, we have clarified that the term co-occurrence in both the abstract and main text, as extreme events occurring in the same location within the same year. We have also updated all the terminology of these co-occurring climate extremes in the main text in order to be consistent (see main text for modifications to the term 'co-occurring' as per the modified text convention here).

We have modified the title as follows:

> The perfect storm? Co-occurring climate extremes in East Africa

Additionally, we have modified the abstract as shown below:

> **Abstract.** Co-occurring extreme climate events exacerbate adverse impacts on humans, the economy, and the environment relative to extremes occurring in isolation. While changes in the frequency of individual extreme events have been researched extensively, changes in their interactions, dependence and joint occurrence have received far less attention, particularly in the East African region. Here, we analyse the joint occurrence of pairs of the following extremes within the same year over East Africa: river floods, droughts, heatwaves, crop failures, wildfires and tropical cyclones. We analyse their co-occurrence on a yearly timescale because some of the climate extremes we consider play out over timescales up to several months. We use bias-adjusted impact simulations under past and future climate conditions from the Inter-Sectoral Impact Model Intercomparison Project (ISIMIP). We find an increase in the area affected by pairs of these extreme events, with the strongest increases for joint heatwaves & wildfires (+940% by the end of the century under RCP6.0 relative to present day), followed by river floods & heatwaves (+900%) and river floods & wildfires (+250%). The projected increase in joint occurrences typically outweighs historical increases even under an aggressive mitigation scenario (RCP2.6). We illustrate that the changes in the joint occurrences are often driven by increases in the probability of one of the events within the pairs, for instance heatwaves. The most affected locations in the East Africa region by these co-occurring events are areas close to the River Nile and parts of the Congo basin. Our results overall highlight that co-occurring extremes will become the norm rather than the exception in East Africa, even under low-end warming scenarios.

Regarding the determination of the main drivers of changes in the occurrence of co-occurring extremes, we consider six extreme events which each have different meteorological and physical drivers. Additionally, we utilize extreme event data from processed impact model simulations, in turn driven by multiple GCMs and scenarios. Given the wealth of data underlying our analysis, diving into the meteorological and physical drivers of the concurrent extreme events presents near-insurmountable challenges. An alternative choice could have been to restrict ourselves to a single impact model and GCM and diagnose the physical drivers, at the cost of increasing hugely the model dependence of our study. A similar choice is often made in e.g. climate modelling studies, with some studies deciding to focus on a smaller set of models (or even a single model) to conduct detailed process studies and others conducting statistical studies on large multi-model ensembles. We nonetheless believe that our analysis does provide interpretable results . This is because: (i) The models only account for climate-induced changes in the hazards and not changes due to land-use change (e.g., deforestation fires) (Lange et al., 2020). Therefore, assuming that our analysis period is sufficiently long to subdue natural variability, we can consider **climate change/global warming** as the only driver of the changes in extreme event occurrence, **and in turn the changes in concurrent extremes**. (ii) We furthermore go deeper in our analysis by splitting the change in concurrent extreme event occurrence into: changes in only one variable per pair, and the **changes in the (coupling) dependence** between extreme events in a pair, as illustrated in Section 3.3. This analysis therefore leads us to figures such as Fig. 5 that illustrate determinants of change in

the concurrent extreme event occurrence as described in the caption. Finally, we note that "dependence" is widely used in a statistical sense, and does not presuppose knowledge of an underlying physical mechanism nor of causality.

We recognize that the use of "driver" that we make in the text can be misleading, as it may be interpreted as "physical" or "climate" driver, while we take a statistical approach in analyzing our data. We have therefore revised our use of the term and now instead refer to "**determinants**" of co-occurring extremes influencing the changes in their cooccurrence. In the revised manuscript, we improved the communication of the method that we use to identify the determinants of changes in the co-occurring extreme occurrences, as well as definition of dependence, by adding the following text in the methods section (Sect. 3.3, Line 131-139) to clarify this point.

> Considering that the processed extreme event simulations account only for climate-induced changes in the extremes (as defined by Lange et al. (2020)), and not for other changes such as land-use, here we only analyse the climate change-driven effects on co-occurring extremes. At a given location, from a statistical perspective, the probability of concurrent extreme events can be affected by the effect of climate change on: (i) the probability of the individual extreme events and/or (ii) the dependence between the events (Bevacqua et al., 2020; Zscheischler et al., 2020b). To gain insights into the determinants of the changes, we compute the change in the probability of concurrent extreme events when assuming: (i) changes in the probability of extreme events in one variable only; and (ii) changes in the coupling (dependence) between the variables only (Bevacqua et al., 2020). Here the term 'dependence' is used in a statistical sense, and does not presuppose knowledge of an underlying physical mechanism nor of causality.

Additionally, we updated the discussion section 5.3 of the manuscript regarding the determinants of co-occurrent extremes (Line 304 - 311) as shown below.

> It is important to note that the six extreme events in this study each have different meteorological and physical drivers, i.e. heatwaves and tropical cyclones have mainly meteorological drivers, while river floods, crop failures, droughts and wildfires have mostly bio-physical drivers. Additionally, some droughts and wildfires are also meteorology-driven. Given that we utilize extreme event data from processed impact model simulations, diving into the meteorological and physical drivers of the concurrent extreme events presents near-insurmountable challenges. Therefore, here we focus on the statistical determinants leading to co-occurring extremes in the same year in East Africa. Here, we consider the changes in the frequency of the individual extremes and their dependence per pair, under a future warmer climate scenario in comparison to the early-industrial period.

To address your concerns, we have made revisions in the manuscript (as shown above) to provide more clarity about our choice of methods and interpretation of the results. We hope that these revisions address your concerns.

**Minor comments**:

1. Lines 74-75 ".. we only identify concurrent extremes .. driven by the same GCM": Please elaborate the calculation method a bit more in detail. Does one GCM have many ensemble members? Then, how are they treated?

**Response:**

The database we use consists of multiple GCMs and impact models. Each GCM includes one (bias-adjusted) ensemble member, as per the ISIMIP2b simulation protocol. The decision to consider only one ensemble member per GCM is a central feature of the ISIMIP protocol and informed by the computational resource limitations of the modelling teams participating in ISIMIP.

By the above text, we mean that for each pair of concurrent extremes, we consider cross-category impact models (as illustrated in Table 1) forced by output from the same GCM, per run, to determine the co-occurrence of these extreme events. For instance, in diagnosing concurrent river floods and wildfires, we use output data from the impact model CLM45 for river floods and data from CARAIB for wildfires where both are driven by the same GCM e.g. GFDL-ESM2M. We then repeat the calculation for the same two impact models but driven both by another GCM, and so on. We finally combine the results after having computed concurrence for all GCMs. As elaborated in lines 74-80 of the manuscript, this is to ensure physical consistency in the analysis. Indeed, this approach avoids that in a given year, one type of extreme event (e.g. Tropical Cyclones) simulated under IOD-positive conditions is compared against another type of extreme event (e.g. soil moisture drought) simulated under IOD-negative conditions.

We have updated the manuscript in lines 74-80 to elaborate the calculation method more, as shown below.

> In this study, a multi-model ensemble approach is followed such that all available impact models per extreme event category (see Table 1), driven by the four aforementioned GCMs, are used to represent the region's exposure to extremes. To guarantee physical consistency in our analyses, we only identify co-occurring extremes from cross-category impact models driven by the same GCM. For instance, in diagnosing co-occurring river floods & wildfires, we use output data from the impact model CLM45 for river floods and data from CARAIB for wildfires where both are driven by the same GCM e.g. GFDL-ESM2M. We then repeat the calculation for the same two impact models but driven both by another GCM, and so on. We finally combine the results after having computed concurrence for all GCMs.

2. Please provide further details on definitions of extreme events. For heatwaves, explain HWMId. For droughts, provide information about considered soil layer depths. For tropical cyclones, does this include all categories and treat them the same?

**Response:**

For the following extreme events, we propose the following additions to their definitions in Table 1 of the revised manuscript, building upon the definitions from Lange et al. (2020).

- **Heatwaves:** Occurrence in entire pixel when the Heat Wave Magnitude Index daily (HWMId; Russo et al., 2017) recorded that year exceeds the 99th percentile of the HWMId during pre-industrial times. Russo et al. (2017) defines HWMId as the annual maximum magnitude of heatwaves, whereby a heatwave consists of a minimum of three consecutive days with temperatures above the daily threshold between 1981 and 2010.

- **Droughts**: Drop of soil water content below the 2.5$^{th}$ percentile of the distribution during pre-industrial times considering periods longer than 6 months. Here, data on monthly soil moisture at different soil layer depths (as close as possible to 100 cm) depending on the impact model was used (See Text S2 in Lange et al. (2020)).
- **Tropical cyclones**: Exposure to hurricane-induced winds (wind speed ≥ 64 knots) sustained for at least one minute during the year
* * *
2. In addition to Table 2, I recommend illustrating the mean and variance in a figure, for instance, using boxplots or probability distribution plots. This way, readers would easily notice the changes and their significance.

**Response:**

We agree. The mean and variance of these individual extreme events (as seen in Table 2) are already illustrated using probability distribution plots in Figures 5 and C1-C4. These figures illustrate the marginal distributions of each extreme event (per scenario), based on the Kernel density estimation (KDE) method (Weglarczyk, 2018), along the top and right axes of the plots for each extreme event. **Additionally**, in the updated manuscript, we now provide boxplots showing the mean percentage areas affected by the individual extreme events under all the climate scenarios (Fig. 4).
* * *
3. In Table 2, are the changes in wildfires statistically significant?

**Response**

No, the changes in wildfires are not statistically significant. In the updated manuscript, we have added an asterisk to the mean values for percentage area affected by extreme events to represent those that are statistically significant in comparison to the early industrial period. See the updated Table 2 and caption below.

- Modified Table 2

**Table 2.** Mean and variance of percentage area affected by extreme events under different climate scenarios.

| Extreme Event | Mean (%) | | | | | Variance | | | | |
|---|---|---|---|---|---|---|---|---|---|---|
| | EI | PD | RCP2.6 | RCP6.0 | RCP8.5 | EI | PD | RCP2.6 | RCP6.0 | RCP8.5 |
| River floods | 1 | 2* | 4* | 7* | 11* | 3 | 5 | 19 | 37 | 72 |
| Heatwaves | 1 | 8* | 53* | 73* | 87* | 8 | 144 | 501 | 460 | 201 |
| Droughts | 2 | 4* | 4* | 5* | 4* | 5 | 14 | 14 | 17 | 1 |
| Crop failures | 0.5 | 1* | 3* | 4.1* | NA | 0.3 | 2 | 4 | 8 | NA |
| Wildfires | 40 | 39 | 39 | 40 | 48* | 275 | 267 | 245 | 265 | 103 |
| Tropical Cyclones | 1 | 1 | 2* | 2* | 2* | 1 | 1 | 1 | 1 | 1 |

Where: EI is the early-industrial period and PD is present-day conditions. Note: (1) The extreme events dataset used in this research does not contain crop failures projections under RCP8.5. (2) We considering a multi-model ensemble approach, whereby the values of mean and variance are averaged across all impact pairs driven by the same GCM. (3) The mean values denoted by an asterisk(*) represent instances where there is a statistically significant difference between the mean percentage area affected by the respective extreme event during that climate scenario, and that during the early-industrial period. Here, we use Welch's $t$ test to determine significant difference in the means (Welch, 1947).

4. Figure 4, Figure 6, and relevant Appendix figures: Please use more distinguishable colors.

**Response:**

For figures 5 and 7 (originally Fig. 4 and 6, respectively) and C1-C4, we now adopt a new color scheme in the updated manuscript with more distinguishable colors as suggested. Additionally, we use this updated color scheme for Fig 1. For illustration purposes, see Fig 5a below:

- Modified Fig 5a

[Figure]

*Figure 5a: Bivariate distribution of river floods & wildfires across 50-year time periods representing the early-industrial period (1861-1910), the present day (1956-2005), and the end of century (2050-2099) under RCP2.6, 6.0, and 8.5. The marginal distributions of each extreme event (per scenario), based on the KDE method (Weglarczyk, 2018), are shown along the top and right axes of the plots. The contours (dotted lines) illustrate smooth estimates of the underlying distribution of co-occurring extremes. The 68th percentile contour, which envelopes data within one standard deviation to either side of the mean, per scenario is used to show a generalized view of the distribution of the percentage of area affected by co-occurring extremes per year during the respective scenarios.*

5. Similar to Figure 1, please add a figure of boxplots for individual events.

**Response:**

Thank you for this suggestion. In the updated manuscript, we have added a figure of boxplots (Fig. 4) showing the annual average percentage of the region affected by the six individual extreme events under past, present and future climate (similar to Fig. 1).

[Figure]

*Figure 4: Boxplots showing the annual average percentage of the region affected by each of the six individual extreme events under past, present and future climates. The extreme events are: RF = River floods, WF = Wildfires, HW = Heatwaves, CF = Crop failures, DR = Droughts and TC = Tropical cyclones. 50-year periods are considered for computing the average for each time window (1861-1910 for early-industrial, 1956-2005 for present day, and 2050-2099 for the future periods). A multi-model ensemble mean is shown that considers all available impact model simulations and driving GCMs in the dataset.. Boxplots display the median (centre line) and upper and lower quartiles (box limits), with whiskers extending to the last value located within a distance of 1.5 times the interquartile range. The yellow circles show mean values. Outliers are not shown. Average global warming level (shown in the brackets within the legend) for each climate scenario, with respect to the early-industrial period, is determined using ISIMIP Global Mean Temperature (GMT) anomalies considering the mean across the respective 50-year windows.*

6. Lines 240-241: There is no information on "global warming level" in the paper. I suggest displaying values of increases in global mean temperature and in the study-area mean temperature next to figure labels in Figure 1.

**Response:**

Thank you for this suggestion. In the updated manuscript, we have included the average global warming level, with respect to the early-industrial period, for the present day and the three future climate scenarios; RCP2.6, 6.0 and 8.5, in Fig. 1 and Fig. 4.

- Modified Fig.1

[Figure]

*Figure 1: Boxplots showing the annual average percentage of the region affected by each of the 15 pairs of concurrent extreme events under past, present and future climates. The extreme events are: RF = River floods, WF = Wildfires, HW = Heatwaves, CF = Crop failures, DR = Droughts and TC = Tropical cyclones. 50-year periods are considered for computing the average for each time window (1861-1910 for early-industrial, 1956-2005 for present day, and 2050-2099 for the future periods). A multi-model ensemble mean is shown that considers all available combinations of impact simulations in the dataset driven by the same GCM. Boxplots display the median (centre line) and upper and lower quartiles (box limits), with whiskers extending to the last value located within a distance of 1.5 times the interquartile range. The yellow circles show mean values. Outliers are not shown. Average global warming level (shown in the brackets within the legend) for each climate scenario, with respect to the early-industrial period, is determined using ISIMIP Global Mean Temperature (GMT) anomalies considering the mean across the respective 50-year windows*

7. Regarding Figure 5: Do we even understand the coupling between two variables of river floods and wildfires in present-day climate? Similar to the panel display in Figure 5, please provide all corresponding maps of actual values in the early-industrial or present-day conditions. Also, explain why some areas have D<1 while other areas have D>1 in Figures 5c and 5i.

**Response:**

Thank you for this comment. In our analysis, we find that dependence of the extreme events is not the main determinant of co-occurrence in the pairs of extremes under future warmer climate; instead, the increase in frequency of these individual extremes is main determinant. For illustration purposes, the figure below (attached to our response) shows a more extensive scale of Fig.6a-c (originally Fig. 5). Here, we observe that for co-occurring river floods & wildfires, the contributing probability ratio (PR) considering coupling of the two events, PR $_{change\ in\ D}$ ≈ 1 (no substantial change) in many locations and in other locations PR $_{change\ in\ D}$ < 1 (less likely to occur). This suggests that dependence of these two extreme events is not the main determinant of co-occurrence, under RCP8.5. This is also the case under present-day conditions (Fig. D5 a-c). As suggested, in the updated manuscript we also provide similar maps showing the contributing probability ratios (PRs) to the change in probability of joint occurrence of different pairs of extreme events under present-day climate (Fig. D5-D9).

[Figure]

*Illustration figure: Contributing PRs to the change in probability of joint occurrence of river floods & wildfires, whereby we illustrate the PR assuming only changes in one of the extremes per pair (first and second columns) and the PR assuming changes only in the dependence of two concurrent extremes (third column). The resulting PRs compare the end-of-century conditions under RCP8.5 to the early-industrial period conditions, whereby PR ≥ 1 represents more likely occurrence of the extremes and P R < 1 represents less likely occurrence. The grey shaded areas did not experience either of the extreme events in each pair (first two columns) or co-occurrences (third column) during the early-industrial period. [**Here we show a more extensive scale than in the manuscript to show probability ratio values with no substantial change i.e. PR ~1 (between 1 and 1.5)** ]*

Additionally, in the updated the manuscript we added a new section i.e., ***Section 5.4: Potential mechanisms underlying the co-occurring extreme events*** (lines 338-348). Here, we provide scientific hypotheses about the possible physical mechanisms leading to some of these co-occurring extreme events and their dependence.

**5.4    Potential mechanisms underlying the co-occurring extreme events**

[revised manuscript text omitted]

---

## Author Response (AR3)

**Response to Editor's comments**:

> Dear Muheki,
>
> Thank you for submitting a revised manuscript.
>
> The reviewer recommends to accept the manuscript.
>
> After a final reading myself, I suggest that you should make clear in the manuscript that by "co-occurring" you mean co-occurring in the same location, and not time (at least that your data set does not allow to make any statements about temporal co-occurrence.
>
> After making this revision please submit also a manuscript version with track changes.
>
> Best regards,
>
> Christian Franzke

We thank the editor for their time and valuable suggestions to improve the manuscript. Below, we illustrate modifications to the manuscript to accommodate the concern raised above. We believe that the manuscript has benefited from these modifications. The following convention is used in this document to illustrate the text modifications in the original manuscript: modified text.

In the abstract, we ensured to clear communicate that these co-occurring extremes occur within the same location and calendar year:

> **Abstract.** Co-occurring extreme climate events exacerbate adverse impacts on humans, the economy, and the environment relative to extremes occurring in isolation. While changes in the frequency of individual extreme events have been researched extensively, changes in their interactions, dependence and joint occurrence have received far less attention, particularly in the East African region. Here, we analyse the joint occurrence of pairs of the following extremes within the same location and calendar year over East Africa: river floods, droughts, heatwaves, crop failures, wildfires and tropical cyclones. We analyse their co-occurrence on a yearly timescale because some of the climate extremes we consider play out over timescales up to several months. We use bias-adjusted impact simulations under past and future climate conditions from the Inter-Sectoral Impact Model Intercomparison Project (ISIMIP). We find an increase in the area affected by pairs of these extreme events, with the strongest increases for joint heatwaves & wildfires (+940% by the end of the century under RCP6.0 relative to present day), followed by river floods & heatwaves (+900%) and river floods & wildfires (+250%). The projected increase in joint occurrences typically outweighs historical increases even under an aggressive mitigation scenario (RCP2.6). We illustrate that the changes in the joint occurrences are often driven by increases in the probability of one of the events within the pairs, for instance heatwaves. The most affected locations in the East Africa region by these co-occurring events are areas close to the River Nile and parts of the Congo basin. Our results overall highlight that co-occurring extremes will become the norm rather 15 than the exception in East Africa, even under low-end warming scenarios.

In the introduction (Line 57-58), we also further clarify that these co-occurring extremes occur within the same location and calendar year:

In this study, we aim to understand the occurrence of compound extremes in East Africa at annual time scales, and focus specifically on co-occurring extremes. Here, the term co-occurring extremes refers to two extreme events occurring within the same location and calendar year. We consider the occurrence of two out of six categories of extreme events within the same year in East Africa namely: river floods, droughts, heatwaves, crop failures, wildfires, and tropical cyclones.